# Hepatitis B virus cccDNA is formed through distinct repair processes of each strand

Lei Wei [1] & Alexander Ploss [1✉]

Hepatitis B virus (HBV) is a highly contagious pathogen that afflicts over a third of the world's population, resulting in close to a million deaths annually. The formation and persistence of the HBV covalently closed circular DNA (cccDNA) is the root cause of HBV chronicity. However, the detailed molecular mechanism of cccDNA formation from relaxed circular DNA (rcDNA) remains opaque. Here we show that the minus and plus-strand lesions of HBV rcDNA require different sets of human repair factors in biochemical repair systems. We demonstrate that the plus-strand repair resembles DNA lagging strand synthesis, and requires proliferating cell nuclear antigen (PCNA), the replication factor C (RFC) complex, DNA polymerase delta (POLδ), flap endonuclease 1 (FEN-1), and DNA ligase 1 (LIG1). Only FEN-1 and LIG1 are required for the repair of the minus strand. Our findings provide a detailed mechanistic view of how HBV rcDNA is repaired to form cccDNA in biochemical repair systems.

---

[1] Department of Molecular Biology, Lewis Thomas Laboratory, Princeton University, Washington Road, Princeton, NJ, USA. ✉email: aploss@princeton.edu

An estimated two billion people have been exposed to Hepatitis B virus (HBV), which has resulted in at least 257 million chronically infected patients. Chronic HBV infection frequently progresses to severe liver disease including fibrosis, cirrhosis, and hepatocellular carcinoma, and results in 880,000 deaths each year[1–8]. HBV belongs to the *Hepadnaviridae* family and its virion contains a compact, partially double-stranded, 3.2 kb relaxed circular DNA (rcDNA) genome with four lesions: on the 5′-end of the minus strand, the covalently linked HBV polymerase and a 10 nucleotide (nt) DNA flap; on the plus-strand, a 5′-capped RNA primer and single-stranded DNA (ssDNA) gap[9,10] (Fig. 1a). Following viral entry mediated by the bile acid transporter NTCP[11], the viral nucleocapsid harboring HBV rcDNA is transported to the nucleus, where the rcDNA is released, and the four lesions on the rcDNA are fully repaired to form cccDNA. cccDNA serves as the template for all HBV viral transcripts and enables chronicity[5]. Current HBV therapies rarely achieve a cure in chronic HBV patients owing to the refractory nature of the stable cccDNA, and blocking cccDNA formation and eliminating existing cccDNA pools are widely regarded as crucial for curing patients[7]. Thus, the elucidation of mechanisms involved in repair of rcDNA to form cccDNA are of utmost importance to determine new drug targets[7].

Viral factors coded by HBV have been shown to be dispensable for the repair of rcDNA to form cccDNA[12,13], and host repair factors are implicated in this process[2,5,12,14–19]. The inability to reconstitute biochemically rcDNA to cccDNA conversion has been a major obstacle for elucidation of the factors and mechanisms governing this critical step in the HBV life-cycle. We previously established a biochemical reconstitution system, and identified five human factors—proliferating cell nuclear antigen (PCNA), the replication factor C (RFC) complex, DNA polymerase delta (POLδ), flap endonuclease 1 (FEN-1), and DNA ligase 1 (LIG1)—that are core components of Okazaki fragment synthesis as a minimal set of factors essential for repairing HBV rcDNA to form HBV cccDNA[20]. However, the detailed molecular mechanisms by which these factors repair the four lesions on HBV rcDNA have not been determined. It is not known whether the repair of lesions on the plus and minus strands of HBV rcDNA require different sets of factors, whether the repair of both strands is dependent on each other, and what the repair steps and kinetics involved in repair of all the lesions are. The main obstacles to answering these questions are: (i) HBV rcDNA substrates harboring minus or plus-strand-specific lesions have not been generated before; and (ii) the repair process and kinetics of all four lesions on HBV rcDNA have not been successfully examined simultaneously before. Here, we generated recombinant HBV rcDNA substrates with strand-specific lesions and deciphered the factors responsible for repairing each strand. We also developed a novel strategy for monitoring the repair process of all

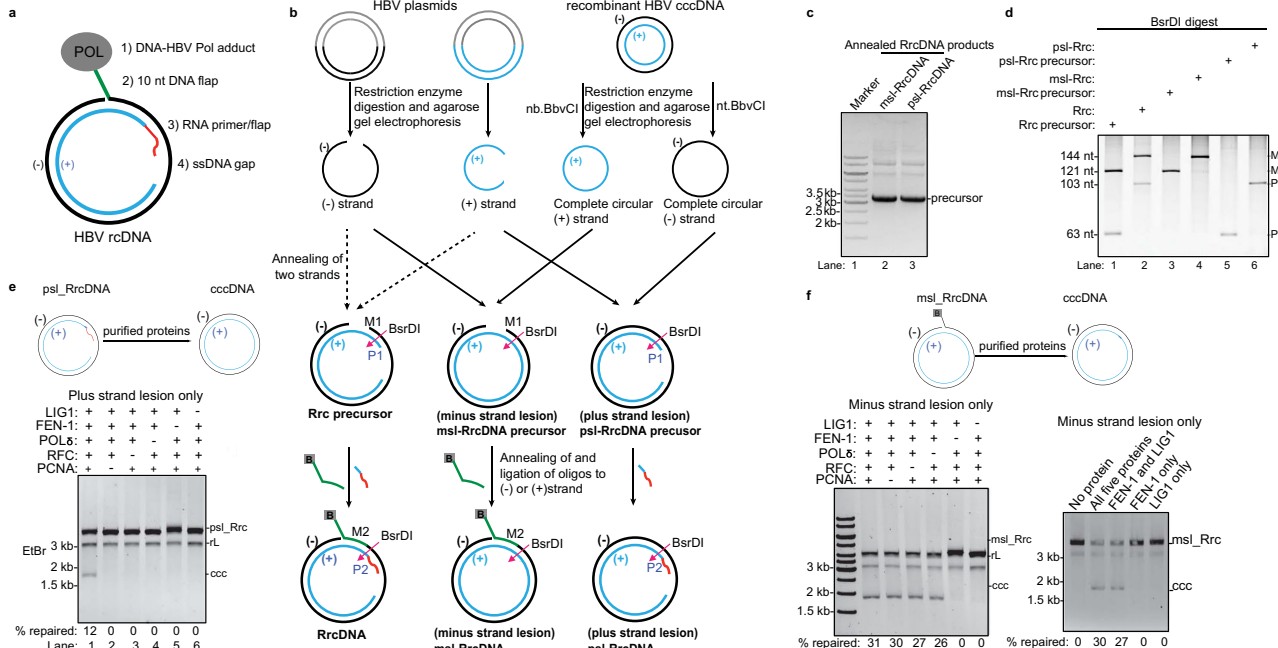

**Fig. 1 Repair of HBV lesions on plus- and minus-strands require different sets of protein factors. a** Characteristics of HBV rcDNA structure. **b** Schematic representation of the generation of recombinant HBV rcDNA (RrcDNA) substrates that contain all lesions (RrcDNA) or only lesions on the plus- (psl-RrcDNA) or minus-strands (msl-RrcDNA). HBV plasmids contains a plasmid backbone (gray) and either the minus strand sequence (black) or the plus-strand sequence (blue). BsrDI cleavage was used to monitor the generation of various RrcDNA substrates (BsrDI restriction sites are indicated by magenta arrows); green line, biotinylated flap; B, biotin; red line, RNA primer. Note that M1 and M2 are two oligos released from the minus-strands of RrcDNA precursor and RrcDNA after BsrDI digestion, whereas P1 and P2 are two oligos released from the plus-strands after BsrDI digestion. **c** Annealing products of psl-RrcDNA and msl-RrcDNA precursors were monitored by Sybr Safe staining. **d** Generation of various RrcDNA precursors was analyzed by formation of M2 and P2 oligos from the corresponding M1 and P1 oligos after BsrDI digestion and urea-PAGE gel electrophoresis followed by Sybr Gold staining. Please note that psl-RcDNA and msl-RrcDNA only contain annealed oligos on the plus-strand and minus strand, respectively. Almost complete conversion from P1 to P2 (psl-RcDNA, lanes 5–6), and M1 to M2 (msl-RrcDNA, lanes 3–4) was observed. **e** All five human protein factors are required for repair of the lesions on the plus-strand. Psl-RrcDNA was mixed with combinations of purified proteins, and cccDNA formation was detected on agarose gels containing ethidium bromide (EtBr). Omission of factors is indicated by "−". **f** FEN-1 and LIG1 are necessary and sufficient for repair of lesions on the minus strand. Omission of factors is indicated by "−". The percentage of cccDNA formed (% repaired) was calculated by dividing the intensity of the ccc band by the sum of the intensities of the RrcDNA, linear RrcDNA and cccDNA bands. The absolute values are displayed above each lane number. *Rrc,* RrcDNA; *rL,* recombinant linear RrcDNA; *ccc,* cccDNA. All experiments were repeated twice with the same results. Source data are provided as a Source Data file.

lesions on HBV rcDNA, and demonstrated the detailed repair mechanism of HBV rcDNA with biochemical reconstitution assays.

## Results

**Repair of rcDNA plus and minus-strands requires different sets of factors**. The minus and plus-strands of HBV rcDNA each contains two different lesions. The covalently linked HBV polymerase and a 10 nt DNA flap are on the minus strand, whereas a 5′-capped RNA primer and single-stranded DNA gap are on the plus strand (Fig. 1a). To determine the combination of human factors required for repairing each strand, we generated recombinant rcDNA (RrcDNA) substrates with defined lesions on either the plus or the minus strand—psl-RrcDNA and msl-RrcDNA, respectively (Fig. 1b–d). Conversion of psl-RrcDNA to cccDNA required all five repair components (Fig. 1e), whereas FEN-1 and LIG1 were sufficient for repairing msl-RrcDNA (Fig. 1f). These results indicate that all five factors are essential for the repair of the plus strand, whereas only FEN-1 and LIG1 are required for repairing the minus strand. We next set to understand how each individual lesion is repaired.

**Examining repair of individual lesions simultaneously**. Understanding the repair intermediates and kinetics of a synchronized reaction is key to deciphering the mechanism of cccDNA formation. Therefore, we monitored the repair intermediates of individual lesions on each strand (Fig. 2a, b). Two recombinant substrates were examined, (i) a NeutrAvidin-RrcDNA complex (NA-RrcDNA), which contains a protein adduct on the minus-strand and mimics the authentic HBV rcDNA; and (ii) RrcDNA, which lacks a protein adduct, and serves as a deproteinated repair intermediate mimic[20]. Removal of the covalently attached HBV polymerase is required to repair rcDNA, and the resultant deproteinated rcDNA has been proposed to be a critical repair intermediate, which undergoes subsequent repair processes on both plus and minus strands to form cccDNA[21–24]. These recombinant substrates were incubated with purified recombinant versions of the five factors at concentrations comparable to those of human nuclear extracts (Supplementary Fig. 1) to determine the kinetics and generation of repair intermediates (Fig. 2a). The reaction products containing a mixture of unrepaired substrates, fully repaired products, and repair intermediates are digested to release fragments harboring different lesions. For unrepaired substrates, four characteristic fragments of interest are expected: Pa (82 nt, containing a free 3′-OH end) and Pb (336 nt, containing a 5′ RNA primer), originating from the plus strand; Ma (413 nt, containing the biotinylated 5′ flap) and Mb (101 nt, containing a free 3′-OH end) are derived from the minus strand (Fig. 2a, b). The fates of these four fragments in various repair intermediates could be monitored by Southern blot (Fig. 2a).

**Repair of plus and minus-strands are independent events**. The presence of a protein adduct on the minus-strand significantly decreased the efficiency of cccDNA formation (Fig. 2c, lanes 1–7 vs lanes 8–14). However, irrespective of the status of a protein adduct on the minus-strand, the repair of the plus-strand was consistently 30% completed within 1 min and plateaued at ~80% at 30 min (Fig. 2d, i). In contrast, the protein adduct significantly retarded repair of the minus-strand (Fig. 2f, i). These results indicate that repair of the plus-strand and the minus-strand of recombinant substrates are independent events in vitro.

**Repair of the plus-strand resembles maturation of Okazaki fragments**. The lesions on the plus-strand resemble the Okazaki fragments, and we have shown that five core factors involved in Okazaki fragment synthesis are necessary and sufficient for plus-strand repair (Fig. 1e). Therefore, we hypothesized that plus-strand repair resembles the maturation process of Okazaki fragments (Supplementary Fig. 2a). In this model, the Pa fragment is equivalent to a primer with a free 3′-OH. This primer-template junction can be recognized by the RFC complex, which recruits and loads PCNA[25]. PCNA interacts with POLδ through its PCNA-interacting peptide (PIP) sequence[26] and serves as the processivity factor for POLδ[27]. Therefore, Pa can be elongated by PCNA-POLδ, which slows down as it reaches the 5′ terminus of Pb and gradually displaces the RNA primer, generating an extended Pa fragment (Supplementary Fig. 2a, step +i). These data are consistent with previous reports showing that POLδ slows down when the enzyme complex encounters DNA or RNA–DNA duplexes[28,29]. Displacement of the RNA primer leads to the formation of an RNA flap structure on Pb that can be recognized and processed by FEN-1, leading to shortening of Pb (Supplementary Fig. 2a, step +ii). The processed Pa and Pb fragments are then joined by LIG1 to form the fully repaired plus-strand (Supplementary Fig. 2a, step +iii).

Consistent with this model, we observed that the Pa fragment was elongated to fill the ssDNA gap and formed one predominant band as early as 1 min (Fig. 2d, lanes 2, 9; *), coinciding with Pb shortening (Fig. 2e, lanes 1–2, 8–9). This band diminished at 3 min, concomitant with the disappearance of Pb and increase of the fully repaired plus-strand product (Fig. 2d–e, lanes 3, 10). The plus-strand repair plateaued between 10 and 30 min regardless of the presence of protein adduct (Fig. 2d–e, i). The plus-strand repair intermediates were largely similar when human cell nuclear extracts were used (Fig. 3a–g). However, the nuclear extract was less efficient, reaching only 45% (vs 80%) completion by 30 min (Fig. 3a–c, g), which could be due to inhibitory factors present in the nuclear extract. Of note, prolonged incubation did not further increase cccDNA formation (Supplementary Fig. 3). The reason of why cccDNA formation does not reach completion remains to be determined but cannot simply be explained by exhaustion of repair factors since we previously demonstrated that addition of a fresh dose of nuclear extract at 60 min only marginally increased repair[20]. Possible reasons include: (1) some repair events are reversible and the reaction reaches equilibrium; (2) some nucleases may degrade cccDNA; (3) formation of dead-end repair intermediates that are refractory for further processing into cccDNA.

The extension of the Pa fragment was dependent on RFC, PCNA, and POLδ, as omission of any of these factors abrogated Pa elongation, as well as cccDNA formation (Fig. 4a, b, lanes 2–4, 8–10). The shortening of Pb was dependent on the FEN-1 endonuclease (Fig. 4c, lanes 4–5, 10–11). However, FEN-1 alone was not sufficient to completely remove the RNA primer on Pb (Supplementary Fig. 4a–c). Pb contains a 19 nt RNA primer, of which eight nt do not pair with the minus-strand, thereby forming an RNA flap (Fig. 2b). FEN-1 alone can remove this unpaired RNA flap[30]. However, when RFC complex, PCNA, and POLδ were added to allow extension of Pa, extensive shortening of Pb was evident (Fig. 4c, lanes 6, 12), indicating complete removal of the RNA primer requires its displacement by Pa extension.

These results indicate that the repair of the plus strand is a highly coordinated process; therefore, we sought to test whether and how sequential addition of FEN-1 and RFC-PCNA-POLδ affects repair of the plus strand. The presence of the biotin moiety on the 5′-end of the minus strand enabled us to capture RrcDNA on magnetic streptavidin beads, and to add and wash off protein factors, permitting sequential addition of various protein factors, and examination of the plus strand repair (Supplementary

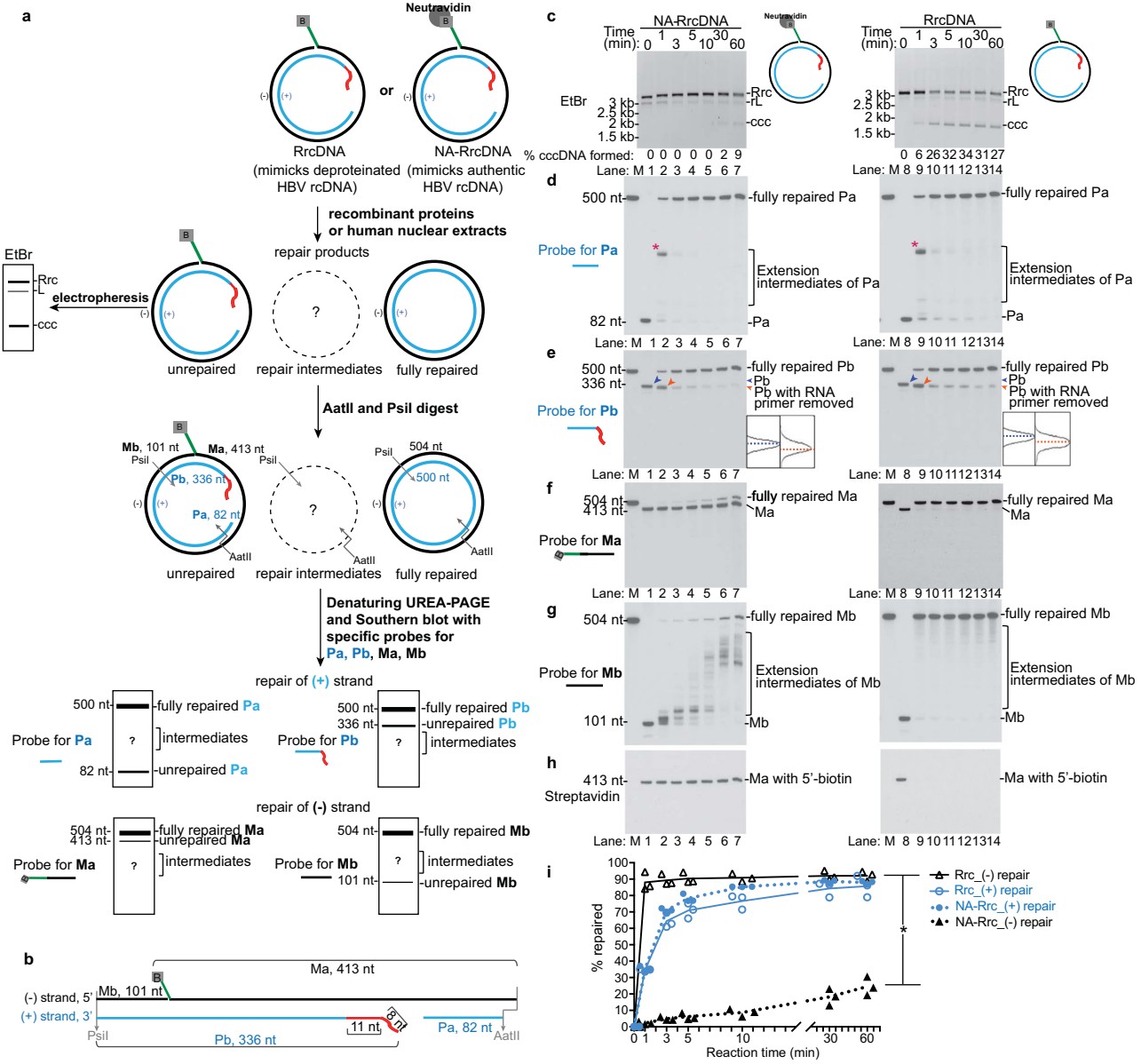

**Fig. 2 Repair of lesions on HBV rcDNA plus- and minus-strands with purified human factors. a** Schematic of examining the kinetics of repair on plus- and minus-strands simultaneously by Southern blot. One half of the repair products were used to monitor cccDNA formation by agarose gel electrophoresis; the other half was digested to generate four fragments of interest: Pa and Pb containing lesions on the plus-strand; Ma and Mb harboring lesions on the minus-strand. These four fragments of repair intermediates (denoted by "?") could be detected by Southern blot as illustrated by idealized blots. The location of the repair intermediates are arbitrarily chosen and indicated by "?". Green line, biotinylated flap; B, biotin; red line, RNA primer. **b** Schematic depicting the four fragments in digested unrepaired RrcDNA. **c** A time course cccDNA formation assay with NeutrAvidin-RrcDNA (NA-RrcDNA, lanes 1–7) or RrcDNA (lanes 8–14) and five purified factors, as depicted in **a**. "% cccDNA formed" was calculated as described in Fig. 1e, f. **d** The repair of the plus-strand Pa fragment is monitored by Southern blot. "*" indicates extension products of Pa that reach the 5′ terminus of Pb. **e** The repair of plus-strand RNA primer-containing Pb fragment is monitored by Southern blot. Traces (plotted with gel-profile function in ImageJ) denote unprocessed (blue arrow head) and processed product (orange arrow head) of Pb fragments, respectively, at 1 min. **f–g** Repair of minus-strand Ma and Mb fragments are monitored by Southern blot. **h** Removal of biotin-containing flap in Ma is detected by Streptavidin blot. **i** The repair efficiency of plus- and minus-strands is calculated from **e** and **f** and plotted. "% repaired" is calculated by dividing the band intensities of fully repaired Pa or Ma by the sum of the band intensities of unrepaired, intermediate, and fully repaired Pa or Ma. *M,* marker; *Rrc,* RrcDNA; *rL,* recombinant linear RrcDNA; *ccc,* cccDNA. All experiments were repeated three times. The lines connect the average value of three measurements at each time point. *P* values are 0.00001, 0.000002, 0.000003, 0.000002, 0.00002, and 0.00005 at time points between 1 and 60 min. Source data are provided as a Source Data file.

Fig. 5a). When all five proteins were added together, repair of the plus strand was efficient at all time points (Supplementary Fig. 5a; 5b–c, lanes 1, 2, 5, 8; and 5d). However, repair of the plus strand was impaired, when FEN-1 was added to the substrates first, washed off, followed by addition of RFC-PCNA-POLδ, and then LIG1 (Supplementary Fig. 5a; 5b–c, lanes 2–3, 5–6, and 8–9; and

5d). Similar results were also observed when RFC-PCNA-POLδ was added before FEN-1 (Supplementary Fig. 5a; 5b–c, lanes 2, 4, 5, 7, 8, 10; and 5d). These results indicate that the sequential addition of these factors led to defective generation of LIG1-ligatable ends in the plus strand, suggesting that these factors function in a concerted fashion in repairing the plus strand.

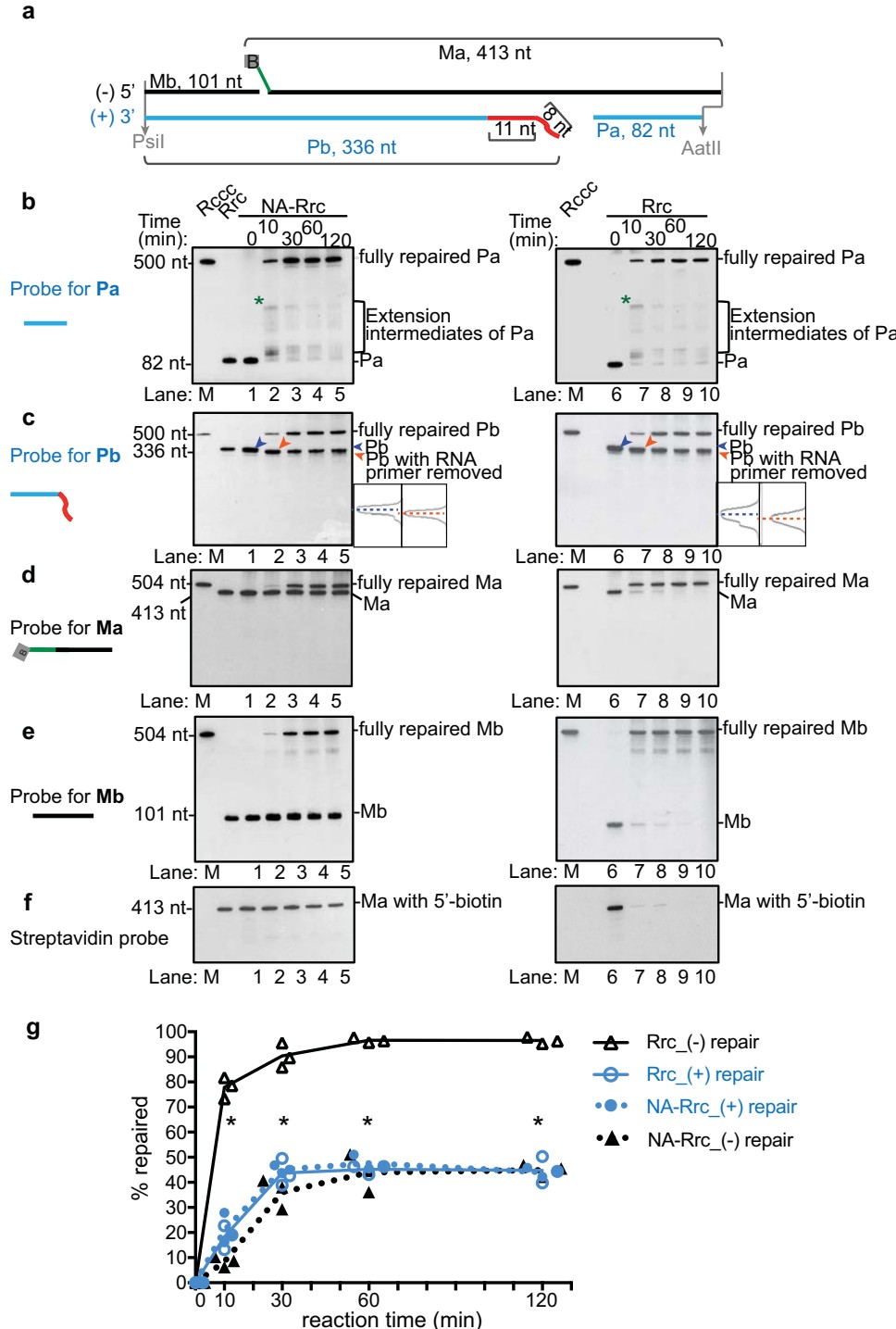

**Fig. 3 Repair of HBV lesions in human nuclear extracts. a** A simplified schematic depicting the four fragments in unrepaired RrcDNA digested by AatII/ PsiI. **b** NA-RrcDNA (lanes 1–5) or RrcDNA (lanes 6–11) was incubated with human nuclear extracts in a time course assay. Repair products at various time points were digested with *AatII/PsiI* and subsequently resolved by UREA-PAGE gel. Repair of the plus-strand Pa fragment was monitored by Southern blot as in Fig. 2d. "*" indicates extended Pa fragments that reach the 5′ terminus of Pa. **c** Repair of the plus-strand RNA primer-containing Pb fragment was monitored by Southern blot as described in Fig. 2e. Traces with dashed blue and orange lines denote unprocessed (blue arrow head) and processed product (orange arrow head) of Pb fragments. **d–e** Repair of the minus-strand Ma and Mb fragments were monitored by Southern blot as in Fig. 2f, g. **f** Removal of the biotin-containing flap of Ma was detected by streptavidin blot as in Fig. 2h. **g** The repair efficiency of plus- and minus-strands is calculated from **c** and **d** and plotted as in Fig. 2i. (−) and (+) repair denote repair of minus and plus-strands, respectively. All experiments were repeated three times, and each individual measurement is plotted. The lines connect the average value of three measurements at each time point. Statistical analyses between the repair efficiencies of the minus strand at each time point are performed by two-stage step-up *t* test method from Graphpad Prism. P values are 0.00001, 0.0003, 0.0003, and 0.000005 at indicated time points (*). Source data are provided as a Source Data file.

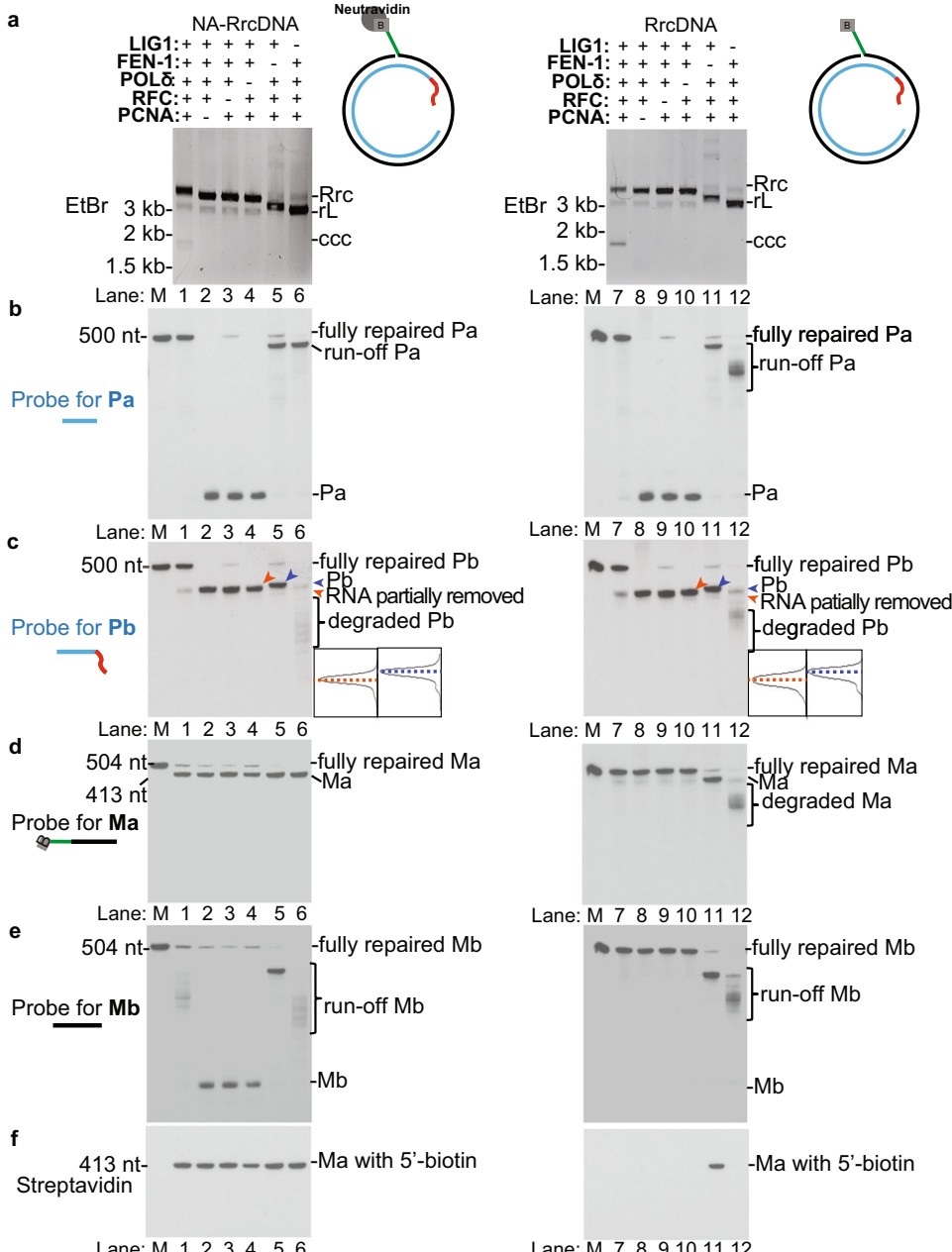

**Fig. 4 Determination of repair intermediates generated when individual human repair factors are omitted. a** cccDNA formation of NA-RrcDNA (lanes 1–6) or RrcDNA (lanes 7–12) by purified human factors was detected by agarose gel electrophoresis containing EtBr. Omission of the factors is indicated by "−". **b** Repair products from **a** were digested with AatII/PsiI and the repair of the plus-strand Pa fragment was monitored by Southern blot as in Fig. 2d. **c** Repair of the RNA primer-containing Pb fragment was detected by Southern blot as in Fig. 2e. Traces with dashed blue and orange line denote unrepaired (blue arrow head) and repaired (orange arrow head) products of Pb fragments. **d–e** Repair of minus-strand Ma and Mb fragments were monitored by Southern blot as in Fig. 2f, g. **f** Removal of the biotin-containing flap of Ma is detected by Streptavidin blot as in Fig. 2h. Rrc, RrcDNA; rL, recombinant linear RrcDNA; ccc, cccDNA. Of note, this figure is related to Supplementary Figure 6, and sections "Omission of individual human factors leads to distinct repair intermediates of the plus-strand" and "Omission of individual human factors leads to distinct repair intermediates of the minus-strand" in the main text. Please refer to these for a comprehensive analysis for the content presented in this figure. All experiments were repeated twice independently. Source data are provided as a Source Data file.

Consistent with our observations, a previous study showed that PCNA/POLδ/FEN-1 cooperatively mediates iterative RNA primer displacement and cleavage cycles in Okazaki fragments, which leads to its complete removal[29].

**Omission of individual human factors leads to distinct repair intermediates of the plus-strand.** We next assessed the function of individual factors in plus-strand repair by examining the

intermediates when specific repair factors were omitted. Consistent with the above-described findings (Fig. 1e), all five factors were required for complete repair of the plus strand (Fig. 4b–c). Omission of RFC, PCNA, or POLδ resulted in the same plus-strand intermediates where Pa was not elongated (Fig. 4b, c, lanes 2–4, 8–10; Supplementary Fig. 6), and the RNA primer on Pb was most likely partially removed by FEN-1, which has been shown to possess this activity (Supplementary Fig. 4c, lanes 1–7). The filling

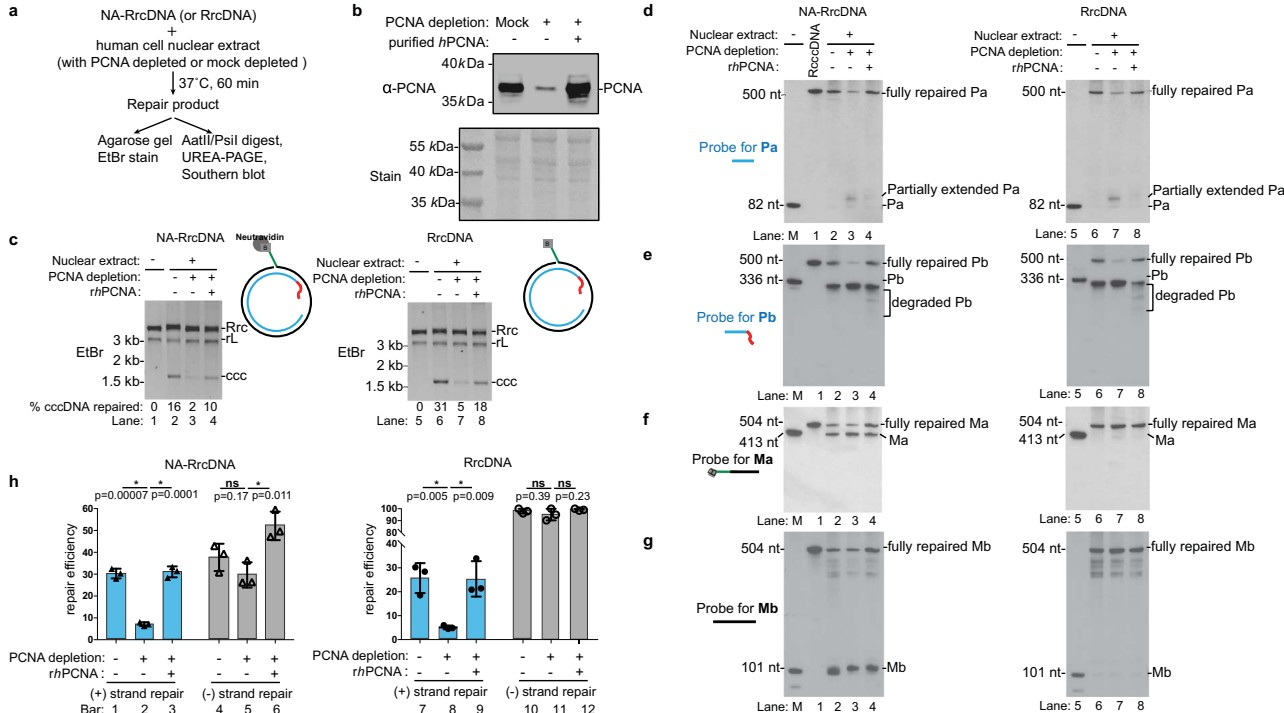

**Fig. 5 Immunodepletion of PCNA from human nuclear extracts diminished plus strand repair. a** Schematic of cccDNA formation assays to test the effects of PCNA depletion in human nuclear extracts on the repair of the plus and the minus strands. **b** PCNA levels in mock-depleted or immunodepleted human nuclear extracts were determined by western blotting. Stain, Ponceau S stain. **c** The depletion of PCNA diminished the cccDNA formation efficiency of NA-RrcDNA (lanes 1–4) and RrcDNA (lanes 5–8) in human nuclear extracts. The percentage of cccDNA formed was calculated as in Fig. 2c. **d–g** Repair of plus-strand fragments Pa, Pb, and minus-strand fragments Ma, Mb was monitored by Southern blot, as shown in Fig. 2d–g. Left panel, repair of NA-RrcDNA; right panel, repair of RrcDNA. **h** Repair efficiency of plus-strand **e** and minus-strand **f** was calculated as in Fig. 2i, and plotted. Statistical analyses between pairs of conditions were performed by two-stage step-up t test method from Graphpad Prism. All experiments were repeated three times. The bar values indicate the average of three measurements and the error bars are the s.d. P values are 0.00007, 0.0001, 0.17, 0.011 for repair of NA-RrcDNA between indicated pairs; whereas P values are 0.005, 0.009, 0.39, and 0.23 for repair of RrcDNA between indicated pairs. Statistical significant p values are indicated by "*". Rrc, RrcDNA; rL, recombinant linear RrcDNA; Rccc, recombinant cccDNA. Source data are provided as a Source Data file.

of the ssDNA gap is thus dependent on RFC, PCNA, and POLδ. The weak band of fully repaired plus-strand when RFC was omitted is most likely due to loading of PCNA independent of RFC on the rare linear RrcDNA substrates (Fig. 4b, c, lanes 3, 9). Similarly, we have shown that immunodepletion of PCNA in human nuclear extracts diminished cccDNA formation[20] (Fig. 5a–c), led to persistent partially extended Pa fragments (Fig. 5d, compare lanes 2 and 3; 6 and 7), impaired repair of the Pb fragment of the plus strand, while the minus strand repair was largely not affected (Fig. 5e–h).

Omission of FEN-1 resulted in unprocessed Pb fragments, and Pa fragments fully extended to the end of the minus-strand Ma template (Fig. 4b, c, lanes 5, 11; Supplementary Fig. 6d, e). This full extension of Pa resulted in linearized RrcDNA, which migrated faster than circular RrcDNA but slower than unrepaired linear RrcDNA (Fig. 4a, lanes 4, 5; 10, 11; Supplementary Fig. 6d, e). In addition, the weak band of fully repaired plus-strand seen when FEN-1 was omitted could be due to a very small proportion of the substrates lacking a flap ligated to the RrcDNA precursor.

Without LIG1, the Pb fragment was degraded and Pa differentially processed, depending on the status of the minus-strand (Fig. 4b, c, lanes 6, 12). As mentioned above, the FEN-1-dependent degradation of Pb was due to its displacement by elongated Pa. When the protein adduct was present, the majority of Pa fragments extended to the end of their template Ma fragment (Fig. 4b, lane 6, run-off Pa; Supplementary Fig. 6f, g), linearizing the repair intermediate (Fig. 4a, lanes 6, 12). However,

without protein adduct, the run-off Pa fragment became shorter and heterogeneous in length (Fig. 4b, lane 12; Supplementary Fig. 6f, g). This is due to Ma, the template for Pa extension, being shortened by FEN-1-dependent degradation when displaced by elongated Mb. Since 5′ protein adduct (such as biotin-streptavidin) reduces FEN-1 activity[31], this degradation is only evident when protein adduct is absent (Fig. 4b, compare lane 6 to lane 12).

**Repair of lesions on the minus-strand.** We next evaluated the repair kinetics of the minus-strand Ma and Mb fragments in our purified protein system (Fig. 2f, g). For the NA-RrcDNA substrate, the Ma fragment containing the flap and protein adduct persisted, and fully repaired minus-strand products only gradually accumulated, reaching <10% at 10 min (Fig. 2f, lanes 1–7; 2i). The slow removal of the flap was also confirmed by the persistence of the biotin moiety at the 5′ end of the flap, which binds to NeutrAvidin (Fig. 2h, lanes 1–7). The slow kinetics of NA-RrcDNA minus-strand repair mirrored those of cccDNA formation (Fig. 2c, lanes 1–7), indicating that minus-strand repair is rate-limiting for cccDNA formation in the presence of protein adduct. Remarkably, without protein adduct, the repair of the minus-strand was more robust than the plus-strand, with close to 100% completion within 1 min (Fig. 2f, g, lanes 8–14; 2i). These results indicate that the slow rate of minus-strand repair is due to the removal rate of 5′ protein adduct.

Like Pa, the Mb fragment contains a free 3′-OH and can be extended by POLδ-PCNA. When FEN-1 was inhibited by the 5′ protein adduct, Mb extension by POLδ-PCNA was evident (Fig. 2g, lanes 1–7). However, when the protein adduct was absent, Mb extension product was minimal and was repaired within 1 min (Fig. 2g, lanes 8–14), indicating removal of the flap and ligation of Ma and Mb were fast and precluded Mb extension.

The repair kinetics with human nuclear extract were comparable to those with purified proteins (Fig. 3d, g), with the exception that aberrant Mb extension was minimal in human nuclear extract when the protein adduct was present. This is conceivably owing to inhibitory factors, such as nucleosomes that restrain POLδ-PCNA–mediated strand displacement[32].

**Omission of individual human factors leads to distinct repair intermediates of the minus-strand.** We next examined the impact on the repair of the minus-strand when individual factors were omitted (Fig. 4d–f). When RFC, PCNA, or POLδ were omitted, the repair of the minus-strand still occurred (Fig. 4d, e, lanes 1–4, 7–10; Supplementary Fig. 6b, c), confirming that these three factors are not required for minus-strand repair.

Omitting FEN-1 drastically diminished the amount of fully repaired minus-strand and led to unprocessed Ma and fully extended Mb fragments with Pa as a template (Fig. 4b, d, e, lanes 5, 11; Supplementary Fig. 6d, e).

When LIG1 was omitted, fully repaired product was not observed, and various repair intermediates accumulated, depending on the presence of 5′ protein adduct (Fig. 4d, e, lanes 6, 12, Supplementary Fig. 6f, g). For the NA-RrcDNA substrate, the majority of Ma was unprocessed since FEN-1 activity was inhibited; Mb was maximally extended to the end of processed/degraded Pa, whose length is heterogeneous (Fig. 4d, e, lanes 6, Supplementary Fig. 6f). On the other hand, for RrcDNA, Ma was shortened due to FEN-1-dependent degradation of displaced Ma, whereas Mb was maximally extended to the end of processed/degraded Pa (Fig. 4d, e, lanes 12, Supplementary Fig. 6g). It is worth noting that when FEN-1 or LIG1 were omitted, the apparent sizes of intermediates, although all were linearized, were smaller with LIG1 than with FEN-1 omission (Fig. 4a, lanes 5–6; 11–12). This was presumably due to the aforementioned FEN-1-mediated shortening of intermediates when LIG1 was omitted (Supplementary Fig. 6f, g).

Collectively, our data show that repair of the minus-strand only requires FEN-1 and LIG1, consistent with a model whereby FEN-1 removes the flap in both NA-RrcDNA and RrcDNA, leaving only a nick in the minus-strand that LIG1 subsequently seals (Supplementary Fig. 2b).

**Aphidicolin treatment specifically delays plus-strand repair.** Our experimental system lends itself to evaluate the precise mechanisms of how small molecules or other factors affect cccDNA formation. Treatment with aphidicolin in our purified protein system led to a delay, but not complete abrogation of cccDNA formation (Fig. 6a–c, i, lanes 1–7 vs 8–14). This delay was owing to specific inhibition of the PCNA-POLδ-mediated elongation of Pa fragments (Fig. 6d, e), whereas minus-strand repair was unaffected (Fig. 6f-i). This observation is also consistent with our findings that plus- and minus-strand repair are independent events. We also have confirmed these findings in human cell nuclear extracts, and found that the defect in the plus strand repair was even more pronounced (Supplementary Fig. 7). Consistent with these findings, aphidicolin also potently inhibited cccDNA formation in hNTCP-HepG2 cells infected with HBV detected by Southern blot and HBeAg (a secreted viral protein) ELISA (Supplementary Fig. 8a, 8b, 8d). Similarly, consistent

with our in vitro finding, another commercially available inhibitor PTPD targeting FEN-1[17,33,34] also diminished cccDNA formation in hNTCP-HepG2 cells infected with HBV in a dose-dependent manner (Supplementary Fig. 8a–c).

**A p21 peptide blocking PCNA and POLδ interaction abrogates plus-strand repair.** As we have shown that PCNA and POLδ are both essential for plus-strand repair and cccDNA formation, we next tested whether a peptide derived from the cyclin-dependent kinase inhibitor p21[WAF1] (KRRQTSMTDFYHSKRRLIFS), which was previously shown to block the interaction between PCNA and POLδ[26,35], could inhibit plus-strand repair and cccDNA formation. We found that this wild-type (WT) p21 peptide completely abolished cccDNA formation in our purified protein system, while a mutant peptide (AAA p21, KRRQTS**A**-T**AA**YHSKRRLIFS) that does not interfere with the interaction of PCNA and POLδ had no effect on cccDNA formation (Supplementary Fig. 9a)[36]. We also confirmed that WT p21 inhibited cccDNA formation in human nuclear extracts in a dose-dependent manner (Supplementary. 9b, c). Most importantly, treatment with 100 μM of WT p21 but not AAA p21 peptide abrogated PCNA-POLδ-mediated elongation of Pa and the repair of the plus-strand, without affecting the repair of the minus-strand (Fig. 6j–p).

**Identification of rate-limiting factors in cccDNA formation.** Finally, we aimed to define the rate-limiting factors in cccDNA formation. For our biochemical assays, we routinely used five repair factors in quantities comparable to those in nuclear extracts[20] (Supplementary Fig. 1). The repair efficiency of rcDNA under such concentrations plateaus, as doubling these concentrations or reducing them by half did not affect cccDNA formation efficiency (Supplementary Fig. 10a). We next aimed to determine the minimal concentration of each factor needed to catalyze the reaction by serially diluting one factor while keeping the other four factors' concentrations the same as their starting concentrations (1.5 μM PCNA, 35 nM RFC, 20 nM POLδ, 100 nM LIG1, and 20 nM FEN-1). The concentration of each factor at which the repair efficiency was reduced to 50% of maximum level (EC50) was determined to be between 0.4 nM (FEN-1) to 20 nM (RFC) (Supplementary Fig. 10b, c). Our results indicate that FEN-1 and RFC are likely to be the least and most rate-limiting factors, respectively, in repairing deproteinated rcDNA intermediates.

## Discussion

A comprehensive understanding of how rcDNA is repaired to form cccDNA is critical for creating new-targeted therapies. We have previously established biochemical systems to fully reconstitute cccDNA formation with purified core components of DNA lagging strand synthesis[20]. These host proteins are essential for cellular replication, and are thus difficult to target in proliferative tissues. However, hepatocytes are mainly quiescent in the steady state[37], and targeted delivery of inhibitory molecules to the liver during short-term treatment may minimize side effects. Thus, such potential therapeutic application may be meritorious to explore further.

The biochemical approaches also open unprecedented opportunities to further delineate the mechanism of rcDNA to cccDNA formation. Here, we have elucidated how these human factors repair each lesion in HBV rcDNA in vitro. The removal of the covalently attached HBV polymerase adduct is required to repair the rcDNA[21–24]. Multiple redundant factors are implicated in removal of HBV protein adduct, these include tyrosyl-DNA phosphodiesterase 2 (TDP2) and FEN-1[15,17,20]. In our biochemical analysis, we utilized NA-RrcDNA and RrcDNA to

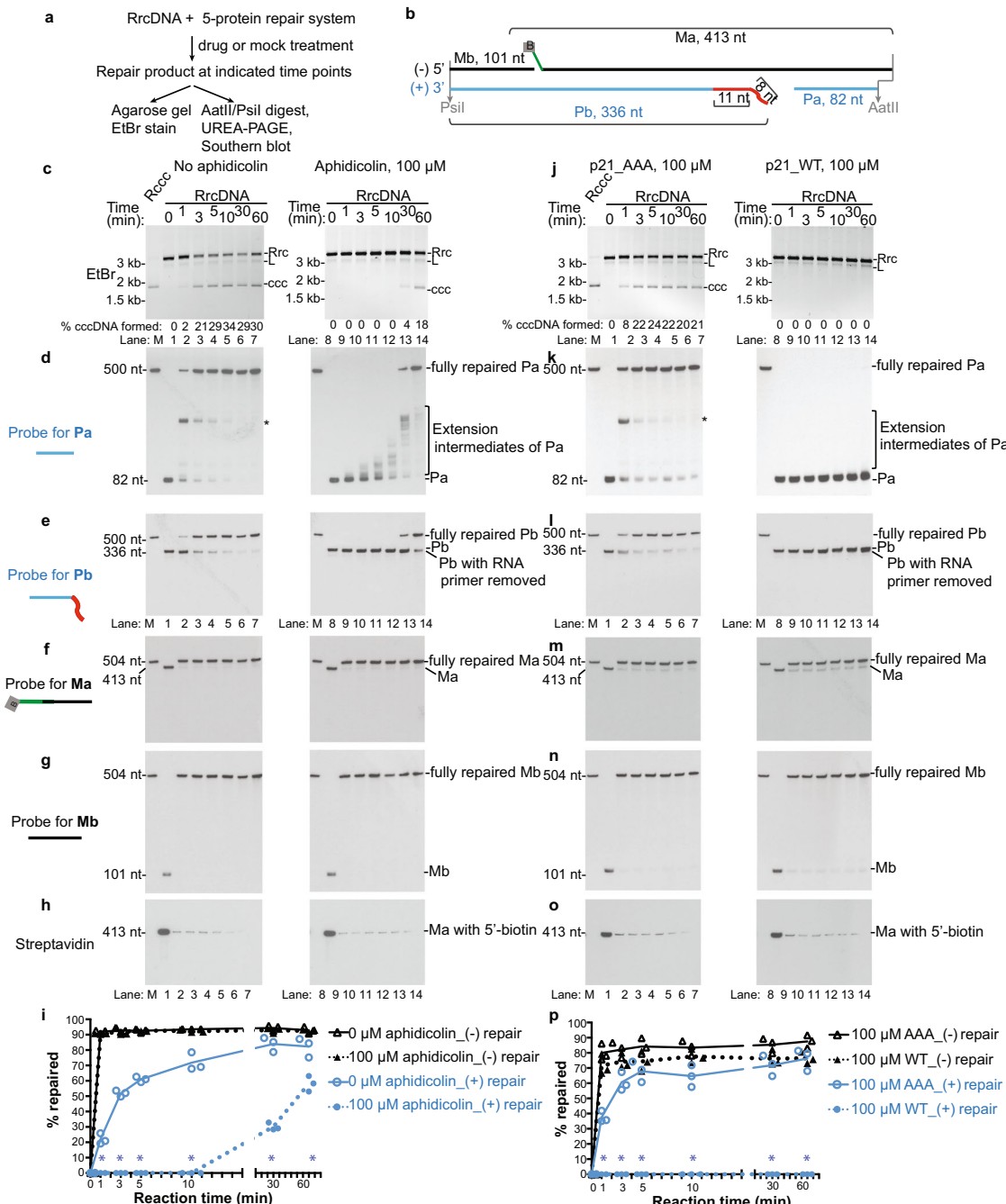

**Fig. 6 Aphidicolin and cyclin-dependent kinase inhibitor p21 peptide specifically inhibit the repair of the rcDNA plus-strand. a** Schematic of time course experiments to test the effects of aphidicolin and p21 peptides on repair of HBV RrcDNA plus- and minus-strand repair. **b** A simplified schematic depicting the four fragments in unrepaired RrcDNA digested by AatII/PsiI. **c** cccDNA formation of the RrcDNA substrate under treatments of mock (1% DMSO, lanes 1–7) or aphidicolin (100 μM in 1% DMSO, lanes 8–14) was detected by EtBr-containing agarose gel. **d–h** Repair of plus-strand fragments Pa, Pb, and minus-strand fragments Ma, Mb was monitored by Southern blot **d–g** or streptavidin blot **h**. **i** Repair efficiency of plus-strand **e** and minus-strand **f** was calculated and plotted as in Fig. 2i. (−) and (+) repair denote repair of minus and plus-strands, respectively. M, marker. **j–p**, same as **c–i**, except that AAA mutant p21 peptide was used as mock treatment, and WT p21 peptide was used as inhibitor to inhibit plus-strand synthesis. All experiments were repeated thrice, and each individual measurement is plotted. The lines connect the average values of three measurements at each time point. Statistical analyses between the repair efficiencies of the minus strand at each time point are performed by two-stage step-up $t$ test method of Benjamini, Krieger, and Yekutieli from Graphpad Prism. $P$ values are 0.0004, 0.000001, 0.0000004, 0.00003, 0.00006, and 0.007 for indicated time points in **i**. $P$ values are 0.00007, 0.00007, 0.00007, 0.0001, 0.00005, and 0.00005 for indicated time points in **p**, and statistically significant $p$ values are indicated by "*". Rrc, RrcDNA; rL, recombinant linear RrcDNA, ccc, cccDNA. Source data are provided as a Source Data file.

mimic authentic HBV rcDNA and deproteinated rcDNA, respectively. The NA-RrcDNA contains a protein adduct non-covalently linked to the minus strand of RrcDNA through NeurAvidin-biotin interaction in lieu of a tyrosyl-phosphodiester bond. Therefore, the effect of TDP2 could not be directly tested in our assays. Another difference between the authentic rcDNA and NA-RrcDNA is that HBV polymerase in the authentic rcDNA is covalently linked to the 5′-end of the minus-strand and occupies the 3′-end of the plus-strand (HBV polymerase can partially fill in the ssDNA gap, but most likely cannot complete the plus strand synthesis), which may impede the completion of repair of both strands. Therefore, the repair of the plus and minus strands in authentic HBV rcDNA can only be completed independently of each other after HBV polymerase removal. In cells infected with HBV, a repair intermediate containing a fully repaired minus-strand has been previously reported[23] and supports this notion. Our data demonstrate that once the protein adduct is removed, the rate of repair of the minus strand is faster than that of the plus strand (Fig. 2d–g, lanes 8–14), which would lead to the formation of a repair intermediate containing a fully repaired minus-strand.

Based on our data and several previous studies, including several reports on repair intermediates in HBV infected cells[12,14–18,23,38], we propose a model for the detailed repair process of HBV rcDNA described in Fig. 7. The removal of the protein adduct can theoretically be carried out by redundant factors: (1) FEN-1 or other nucleases (step i), and (2) TDP2 or proteases (step i′, the resultant deproteinated rcDNA can be mimicked by RrcDNA). After the removal of the protein adduct, minus-strand repair with purified proteins occurs as steps –i and –ii (Fig. 7). As such, FEN-1 and TDP2 may work in concert in cells to remove the protein and flap, as FEN-1 removes the flap with a protein adduct (biotin-NeutrAvidin) very inefficiently (Fig. 2h, Supplementary Fig. 4d, e), thus removal of HBV polymerase by TDP2 would facilitate this process. However, it is conceivable that FEN-1 is the major factor to remove the protein and the flap in cells, as HBV cccDNA formation is inefficient in cell culture, which could be due to FEN-1's low activity on intact rcDNA. The plus-strand repair in vitro occurs in a manner similar to Okazaki fragment synthesis. The 3′-end of the incomplete plus-strand is engaged by PCNA, which is loaded by RFC ( + ii). PCNA recruits POLδ via the PIP on POLδ (+iii), and the PCNA-POLδ complex completes the plus-strand synthesis. This displaces the RNA primer on the 5′-end of the plus-strand (+iv), generating a flap structure that is recognized and removed by FEN-1, leaving a single nick on the plus-strand (+v). The nick is subsequently ligated by LIG1, completing the plus-strand repair (+vi).

The efficiency of repair in human nuclear extract is comparable to that in the purified protein system. It is worth noting, that we observed that PCNA depletion in human nuclear extracts led to a trend of a slight decrease in repair efficiency of the minus strand, albeit not statistically significant (Fig. 5h, bars 4–5). Adding recombinant PCNA back to the depleted extracts led to a statistically significant increase in minus-strand repair (Fig. 5b–h, bars 5–6). These results suggest that PCNA may facilitate the repair of the minus strand in nuclear extracts, although it is not required. There are numerous DNA-binding proteins in nuclear extracts that compete with repair factors for access to the substrates. Since PCNA has been shown to interact with POLδ, FEN-1, and LIG1[26,39], it may have an additional function as a scaffold to recruit these proteins to facilitate the minus strand repair in cells. Thus, it is highly likely that the formation of cccDNA in human hepatocytes infected with HBV is more complex, with more factors that could perform redundant and/or regulatory functions[12,14,18,19]. However, some factors could affect cccDNA formation indirectly, our system also provides a powerful platform to examine if these factors are directly involved in rcDNA repair.

cccDNA biogenesis in human hepatocytes is a multi-step process, which involves viral entry, nucleocapsid transport, capsid disassembly, repair of lesions in rcDNA, and chromatinization of cccDNA. As such, cccDNA biogenesis requires extensive interplay of viral and host factors. Our study showed that the biochemical approach is a powerful tool to reveal the mechanisms of rcDNA repair step. Many questions still remain, how do these repair factors cooperate with each other? What limits the complete conversion of rcDNA to cccDNA? Are the repair kinetics and intermediates similar in cells infected with HBV? The biochemical approach combined with HBV cell culture models can serve as a platform to bridge these knowledge gaps. Meanwhile, similar biochemical approaches could also be used to dissect other steps of cccDNA biogenesis, such as capsid disassembly and cccDNA chromatinization.

## Methods

**Cell lines.** The hNTCP-expressing HepG2 cell clone 3B10[40] was authenticated by its susceptibility to HBV infection, and was maintained in Dulbecco's modified Eagle medium (DMEM; Thermo Fisher Scientific, Waltham, MA) supplemented

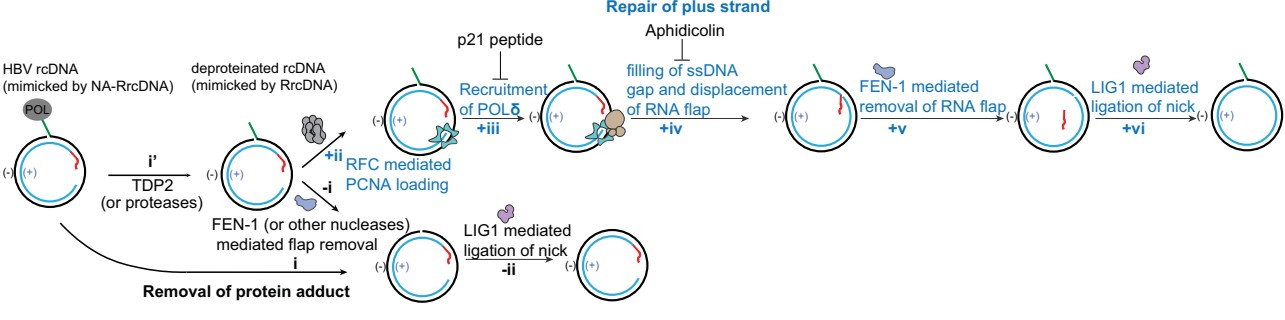

**Fig. 7 A putative model of the repair of HBV rcDNA.** The HBV polymerase adduct is first removed by redundant factors FEN-1 (step i) and TDP2 (step i′). The subsequent repair of the minus- and plus-strands are independent of each other, and the repair of the minus-strand only requires FEN-1 and LIG1 and proceeding, as steps -i and -ii. The repair of the plus-strand resembles Okazaki fragment synthesis, requiring more steps as shown from steps +ii to +vi. First, the RFC complex recognizes the 3′ end of the plus strand and loads PCNA to this primer-template junction (step +ii). PCNA then recruits POLδ via the PCNA-interacting peptide (PIP) sequence on POLδ (step +iii). This step can be specifically inhibited by the p21WAF1 peptide (KRRQTSMTDFYHSKRRLIFS). Subsequently, POLδ is activated by PCNA and fills the ssDNA gap and displaces the RNA primer on the 5′-end of the plus-strand (step +iv). This step can be delayed by aphidicolin treatment. The displaced RNA primer adopts a flap structure that can be recognized and removed by FEN-1, leaving one nick on the plus-strand (step +v). At last, LIG1 ligates this nick and completes the repair of the plus-strand (step +vi).

with 10% (vol/vol) fetal bovine serum (FBS, Hyclone), 100 units/ml of penicillin, and 100 μg/ml streptomycin (Thermo Fisher Scientific, Waltham, MA).

**Generation of recombinant cccDNA by minicircle technology.** Recombinant HBV cccDNA (Rccc) contains a 39 bp insertion (CCCCAACTGGGGTAACCTTT GGGGCTCCCCGGGCGCGACCC) in the polymerase domain between nt 2849 and nt 2850 (nucleotide numbers correspond to those in U95551.1), which does not lead to a frame shift, and the 3221 bp recombinant cccDNA functionally behaves like authentic HBV cccDNA[41]. RcccDNA was generated as previously described[20,41,42]. In brief, the parental Rccc production plasmid (pLW25) containing the HBV genotype D genome (GenBank accession number: U95551.1) was amplified in the DNA methylation deficient *E.coli* strain ZYCY10P3S2T *dam-/ dcm-*, which was subsequently treated with L-arabinose (Sigma Aldrich) at a final concentration of 0.01% (w/vol) to induce the generation of HBV RcccDNA. RcccDNA was further purified by NucleoBond Maxi prep kit (Macherey Nagel).

**Generation of RrcDNA and NeutrAvidin-RrcDNA complex.** The method to generate RrcDNA and NeutrAvidin-RrcDNA complex is depicted in Fig. 1b, and has been previously described in detail[20]. In brief, plasmids pLW213 and pLW227 were used to produce minus and plus-strand ssDNA, respectively.

The minus-strand ssDNA and plus-strand ssDNA were generated by digesting pLW213 and pLW227 with Nt.BspQI and BssHII (NEB), respectively. The digests were denatured in denaturing gel-loading buffer (DS611, BioDynamics Laboratory) at 80°C for 10 min, and ssDNA was subsequently purified by agarose gel electrophoresis and dissolved in RNase-free water. To generate RrcDNA precursor, the minus and plus-strand ssDNAs were mixed at a 1:1 molar ratio in 500 mM NaCl and subsequently annealed and purified by agarose gel electrophoresis. To generate RrcDNA, RrcDNA precursor was incubated and ligated with oligo PU-O-5573 (5′ Biotin-GAAAAAGTTGCATGGTGCTGGTG, Integrated DNA Technologies, Coralville, IA) and oligo PU-O-5670 (5′ rGrCrArArCrUrUrU rUrUrCrArCrCrUrCrUrGrCACGTCGCATGGAGACCACCGT, underlined 11 ribonucleotides anneals to the minus-strand, whereas the eight non-underlined ribonucleotides do not, Integrated DNA Technologies) to restore the flap structure and RNA primer on the minus and plus-strands, respectively. The ligation efficiency was determined as described in Fig. 1b, d. In brief, purified RrcDNA precursor and RrcDNA were digested with BsrDI (NEB). The digest was purified by phenol–chloroform extraction and resolved on a 10% (w/vol) urea-polyacrylamide gel electrophoresis (PAGE), which was subsequently stained with SYBR™ Gold (Thermo Fisher Scientific) to visualize the M1/P1 single-stranded oligos originated from RrcDNA precursor and M2/P2 single-stranded oligos produced by RrcDNA after BsrDI digestion as shown in Figs. 1b, d, lanes 1–2.

To generate the NeutrAvidin-RrcDNA complex, RrcDNA was incubated with NeutrAvidin (Thermo Fisher Scientific) at a final concentration of 4 mg/ml in buffer containing 20 mM HEPES-KOH (pH 8.0) and 50 mM KCl at 37°C for 30 min.

**Generation of RrcDNA substrates that only contain lesions on the minus or plus-strand.** The method to generate RrcDNA substrates that only contain lesions on the minus or plus-strand is depicted in Fig. 1b, d. To generate plus-strand lesion-RrcDNA (psl-RrcDNA), complete circular minus-strand ssDNA was first generated by digesting HBV RcccDNA with nt.BbvCI (NEB), which cleaves the plus-strand and leaves the minus-strand intact. Complete circular minus-strand ssDNA was subsequently separated from the plus-strand by agarose gel electrophoresis, and was dissolved in RNA-free water. Complete circular minus-strand ssDNA was then annealed to plus-strand ssDNA (from pLW227, described above), which led to the formation of psl-RrcDNA precursor (Fig. 1c, lane 3). RNA–DNA hybrid Oligo PU-O-5670 (5′-rGrCrArArCrUrUrUrUrUrCrArCrCrUrCrUrGrCAC GTCGCATGGAGACCACCGT, Integrated DNA Technologies, Coralville, IA) was annealed and ligated to the psl-RrcDNA precursor to restore the RNA primer on the plus-strand ssDNA to generate the psl-RrcDNA. The ligation efficiency was determined by BsrDI digestion as described above and shown in Fig. 1b, d, lanes 5–6.

Minus-strand lesion-RrcDNA (msl-RrcDNA) was generated in a similar manner. Complete circular plus-strand ssDNA was generated by digesting HBV RcccDNA with nb.BbvCI (NEB), which cleaves the minus-strand and leaves the plus-strand intact. Purified complete circular minus-strand ssDNA was then annealed to minus-strand ssDNA (from pLW213, described above), which led to the formation of msl-RrcDNA precursor (Fig. 1c, lane 2). DNA oligo PU-O-5573 (5′ biotin-GAAAAAGTTGCATGGTGCTGGTG, Integrated DNA Technologies, Coralville, IA) was annealed and ligated to the msl-RrcDNA precursor to restore the DNA flap structure on the minus-strand ssDNA to generate the msl-RrcDNA. The ligation efficiency was determined by BsrDI digestion as described above and shown in Fig. 1b, d, lanes 3–4.

**Preparation of human cell nuclear extracts.** Human cell nuclear extracts from human hepatoma cell line HepG2 expressing human NTCP (hNTCP, clone 3B10)[40] were prepared as previously described[20]. In brief, cells were harvested, resuspended in hypotonic buffer (10 mM HEPES-KOH (pH 8.0), 1.5 mM MgCl₂, 10 mM KCl, 1 mM DTT, 0.5 mM phenylmethylsulfonyl fluoride (PMSF) and 1× Protease Inhibitor

Cocktail (Sigma Aldrich) and lysed using a Dounce homogenizer. Cell lysates were stained with Trypan blue (Thermo Fisher Scientific), and were examined under the microscope. More than 90% stained cells indicated successful cell lysis. The lysed cells were then spun at $1500 \times g$ for 5 min. The cell pellet (the nuclei fraction) was resuspended in high salt solution containing 20 mM HEPES (pH 8.0), 1.5 mM MgCl₂, 700 mM KCl, 5% (vol/vol) glycerol, 0.5 mM PMSF, and 1× Protease Inhibitor Cocktail (Sigma Aldrich). The mixture was incubated at 4°C for 30 min before being subjected to centrifugation at $20,000 \times g$ for 30 min. The supernatant was recovered, concentrated, and dialyzed against buffer containing 20 mM HEPES-KOH (pH 8.0), 50 mM KCl, 5% glycerol, 1 mM DTT, and 0.5 mM PMSF. The final protein concentrations of nuclear extracts were 10-20 mg ml⁻¹.

**Recombinant proteins.** Recombinant human PCNA, RFC complex, POLδ complex, LIG1, and FEN-1 used in this study are the same batch used in the previous publication[20].

**cccDNA formation assay in human nuclear extracts.** Various RrcDNA substrates (60 ng, 29 fmol, ~$2 \times 10^{10}$ molecules) were incubated with human nuclear extract in reaction buffer containing 20 mM Tris-HCl (pH 7.6), 50 mM KCl, 0.1 mM dNTP, 5 mM MgCl₂, 1 mM reduced glutathione, 2.6 mM ATP, 26 mM phosphocreatine disodium (Sigma Aldrich), and 6 μg ml⁻¹ creatine phosphokinase (Sigma Aldrich). The mixture was incubated at 37°C, and the reaction was terminated by addition of sodium dodecyl sulfate and ethylenediaminetetraacetic acid (EDTA) to final concentrations of 0.5% (w/vol) and 25 mM at indicated time points, respectively. The solution was subsequently treated with proteinase K for 1 hour, and was then extracted by phenol–chloroform and dissolved in RNAse-free water. To monitor cccDNA formation, the repair products were loaded onto a 0.7% (w/vol) agarose gel containing 0.05 μg ml⁻¹ ethidium bromide and run at 4 V/cm for 2 hours before the gel was visualized on Typhoon™ FLA 9500 (GE Healthcare Lifesciences). The intensity of DNA bands was quantified by ImageJ.

**Reconstitution of cccDNA formation by purified human proteins.** cccDNA formation experiments with purified human factors were performed similarly to those experiments with human nuclear extracts described above. The only exception was that 1.5 μM PCNA, 35 nM RFC, 20 nM POLδ, 100 nM LIG1, and 20 nM FEN-1 (or concentrations as otherwise indicated) were used in place of human nuclear extracts.

**Examination of the repair of individual lesions simultaneously.** The schematic to examine the repair of individual lesions of various HBV RrcDNA substrates was described in Fig. 2a, b. In brief, cccDNA formation assays with human nuclear extracts or purified human proteins were performed as described above. The repair products at various time points were digested with proteinase K, phenol–chloroform purified, and subsequently subjected to restriction enzyme digestion with AatII and PsiI (NEB). The fragments released by the restriction enzyme digestion contain two fragments that harbor plus-strand lesions (Fig. 2a, Pa and Pb) and two fragments that encompass minus-strand lesions (Fig. 2a, Ma and Mb). The digests containing these fragments were denatured at 95°C for 10 min in denaturing buffer containing 95% (vol/vol) formamide, 0.025% (wt/vol) bromophenol blue, 0.025% (wt/vol) xylene cyanol and 5 mM EDTA, before being resolved on a denaturing 5.5% (w/vol) urea-PAGE gel. DNA was subsequently transferred onto a positively charged nylon membrane (Sigma Aldrich, St. Louis, MO), and was cross-linked to the membrane by UV irradiation (Stratalinker 1800, Stratagene) at 120,000 μJ cm⁻², and incubated with pre-hybridization solution supplied in the digoxigenin (DIG) High Prime DNA Labeling and detection Starter Kit II (Sigma Aldrich) for 1 hour at 39°C. The membrane was subsequently hybridized with alkaline-labile DIG-labeled HBV probe specific for Pa fragment (ACGGCAGACGGAGAAGGGGACGAGAGAGTCCCAAGCGACCCCGAG AAGGGTCGTCCGCAGGATTCAGCGCCGACGGGACGTAAACAAAGG) of the plus-strand at 39°C for 14 hours to monitor the repair kinetics and repair intermediates of Pa fragment. The hybridized membrane was washed with washing buffer (0.1 M maleic acid (pH 7.5), 150 mM NaCl, 0.3% (vol/vol) Tween-20). The membrane was then blocked and subsequently incubated with alkaline phosphatase-conjugated DIG antibody at a 1:10,000 dilution. After a 1-hour incubation at room temperature, the membrane was washed twice with washing buffer for 15 min. The membrane was subsequently subjected to chemiluminescent detection by incubation with the substrate and the signal detected on X-ray film (Thermo Fisher). The film was then scanned with an Epson scanner at a resolution of 300 dpi, and the intensities of the bands were quantified with ImageJ version 1.52a.

To monitor the repair of Pb, the membrane was stripped by treatment with 0.4 M NaOH to remove the alkaline-labile DIG probe for Pa, and re-probed with DIG-labeled HBV probe specific for Pb fragment (TAAGGGTCGATGTCCATGCCCC AAAGCCACCCAAGGCACAGCTTGGAGGCTTGAACAGTAGGACATGAAC AAGAGATGATTAGGCAGAGG). Similarly, the membrane was then stripped, and re-probed with DIG-labeled HBV probe specific for Ma (CCTTTGTTTACG TCCCGTCGGCGCTGAATCCTGCGGACGACCCTTCTCGGGGTCGCTTGGG ACTCTCTCGTCCCCTTCTCCGTCTGCCGT) and Mb (CCTCTGCCTAATC

ATCTCTTGTTCATGTCCTACTGTTCAAGCCTCCAAGCTGTGCCTTGGGTGGCTTTGGGGCATGGACATCGACCCTTA).

To monitor the removal of biotin from the Ma fragment, the biotin-containing fragments were detected by the Chemiluminescent Nucleic Acid Detection Module (Thermo Fisher Scientific), according to the manufacturer's instructions. In brief, the membrane was incubated with Blocking Buffer for 15 min. The membrane was subsequently incubated with Streptavidin-Horseradish Peroxidase Conjugate (1:800 dilution) for 20 min, room temperature. After four washes with 1× Wash Buffer, the membrane was incubated with Substrate Equilibration Buffer and was then subjected to chemiluminescent reaction with substrates provided in the kit. The membrane was then exposed to X-ray film (Thermo Fisher Scientific). The developed autoradiography film was then scanned with an Epson scanner at 400 dpi resolution and the bands quantified with ImageJ. Figures 2i, 3g, 6i, 6p were all plotted with Graphpad Prism Software version 7.0d (Graphpad).

**Examination of the removal of DNA and RNA flaps by FEN-1.** To examine of the removal of DNA and RNA flaps by FEN-1 in Supplementary Fig. 4, RrcDNA (60 ng) or NA-RrcDNA (60 ng) was incubated with 20 nM FEN-1 in reaction buffer modified from a previous study[43] consisting of 20 mM Tris-HCl (pH 7.6), 50 mM KCl, 0.1 mM dNTP, 5 mM MgCl2, 1 mM reduced glutathione, 2.6 mM ATP, 26 mM phosphocreatine disodium (Sigma Aldrich), and 6 µg ml−1 creatine phosphokinase (Sigma Aldrich). The reaction was terminated at indicated time point and the repair products purified as described in the cccDNA formation assay. The purified repair products were digested with BsrDI (NEB) and subsequently resolved on a denaturing 8% (w/vol) urea-PAGE gel. The removal of RNA flap from the plus-strand was detected by Southern blot as described above using alkaline-labile DIG-labeled HBV probe specific for Pd fragment (Supplementary Fig. 4b, GAC ATTGCAGAGAGTCCAAGAGTCCTCTTATGTAAGACCTTGGGCAACATT CGGTGGGCGTTCACGGTGGTCTCCATGCGACGTGCAGAG). The membrane was subsequently stripped by 0.4 M NaOH and the removal of DNA flap from the minus-strand was detected by alkaline-labile DIG-labeled HBV probe specific for Md fragment (Supplementary Fig. 4b, AACGACCGACCTTGAGGCATACTTC AAAGACTGTTTGTTTAAAGACTGGGAGGAGTTGGGGGAGGAGATTAGA TTAAAGGTCTTTGTACT).

**Evaluating the effects of sequential addition of protein factors on plus strand repair of recombinant HBV rcDNA (related to Supplementary Fig. 5).** In all, 2 µg of RrcDNA (with a biotin moiety on the 5′ of the minus strand) was immobilized on 60 µl magnetic streptavidin beads (Dynabeads M280, Invitrogen) in binding/wash buffer containing 20 mM HEPES, pH 7.8 @ 4 °C, 1 M NaCl, and 0.5 mM EDTA. After rotation at RT for 30 min, the beads were washed three times with buffer containing 20 mM HEPES, pH 7.8 @ 4 °C, 50 mM KCl, 5% glycerol, and 1 mM DTT at RT. The beads were then divided to Eppendorf tubes, so that each tube contained 200 ng substrates. Indicated protein factors in buffer containing 20 mM Tris-HCl (pH 7.6), 50 mM KCl, 0.1 mM dNTP, 5 mM MgCl2, 1 mM reduced glutathione, 2.6 mM ATP, 26 mM phosphocreatine disodium, and 6 µg ml−1 creatine phosphokinase were added to the beads, which was subsequently incubated for indicated durations (60 min, or 10 min, or 3 min) on a rotator at 37 °C. After incubation, the supernatant was removed, and the beads were washed with binding/wash buffer for three times, then washed with low salt buffer containing 20 mM HEPES, pH 7.8 @ 4 °C, 50 mM KCl, 5% Glycerol, and 1 mM DTT twice. Other indicated protein factors were subsequently added to the beads and incubated as described above. At the end of the reaction, the beads were washed with binding/wash buffer three times, then washed with low salt buffer twice, before being subjected to AatII/PsiI digestion at 37 °C for 2 hours. The resultant digests were denatured at 95 °C for 10 min in denaturing buffer containing 95% (vol/vol) formamide, 0.025% (wt/vol) bromophenol blue, 0.025% (wt/vol) xylene cyanol and 5 mM EDTA. The denatured samples were then resolved on a 5.5% (w/vol) denaturing urea-PAGE gel, and the plus strand repair was monitored via Southern blot as described in section 'Examination of the repair of individual lesions simultaneously'.

**Inhibition experiments with aphidicolin and p21 peptides.** Aphidicolin (Catalog number 102513, VWR) was dissolved in 100% dimethyl sulfoxide (DMSO) (Sigma Aldrich) to a final concentration of 10 mM, which was subsequently diluted to 1 mM in buffer containing 20 mM HEPES-KOH (pH 8.0), 50 mM KCl, 5% glycerol, 1 mM DTT, and 0.5 mM PMSF. In Fig. 6c–h, and Supplementary Fig. 7, aphidicolin was added 20 min prior to addition of substrates to cccDNA formation reactions to achieve a final concentration of 100 µM, 1% DMSO. Control treatments without aphidicolin contained 1% DMSO.

Lyophilized WT p21WAF1 peptide (KRRQTSMTDFYHSKRRLIFS) and a mutant peptide (AAA p21WAF1, KRRQTS̲A̲T̲A̲AYHSKRRLIFS) were synthesized by Genscript. The peptides were dissolved to a final concentration of 5 mM in buffer containing 20 mM HEPES-KOH (pH 8.0), 50 mM KCl, 5% glycerol, 1 mM DTT. 20 min prior to the addition of substrates, the peptide was further diluted to indicated concentrations in cccDNA formation assays. Fig. 6i, 6p, and Supplementary Fig. 9c were plotted with Graphpad Prism Software version 7.0d (Graphpad).

**HBV infection of HepG2-hNTCP cells in the presence of aphidicolin and PTPD.** HBV infection of HepG2 cells expressing hNTCP (HepG2-hNTCP) was performed as previously described[20], with minor modifications. Tissue culture-derived HBV (Genotype D) from HepG2.2.15 cells was used for all experiments. In all, 1.2 × 10^6 HepG2-hNTCP cells/well were plated in a six-well plate, and after an overnight incubation, cells were treated for 12 hours with various concentrations of aphidicolin (VWR, cat# 102513) or FEN-1 inhibitor PTPD (AK Scientific) in the presence of DMEM supplemented with 10% (vol/vol) FBS and 2% (vol/vol) DMSO. The cells were then challenged with HBV virus at a multiplicity of infection (MOI) of 2000 in the presence of aphidicolin or PTPD and DMEM supplemented with 10% (vol/vol) FBS, 4% (w/vol) PEG 8000 (Sigma Aldrich), and 2% (vol/vol) DMSO (Sigma Aldrich). After 18 hours, the inoculum was aspirated and the cells washed three times with DMEM medium. The cells were subsequently maintained in aphidicolin and PTPD supplemented with 10% (vol/vol) FBS and 2% (vol/vol) DMSO. To ensure the efficacy of drug treatments, the media was replaced every 12 hours with fresh drug-containing media until cells were harvested at 60 hours post infection.

**Detection of cccDNA and HBeAg levels in HepG2-hNTCP cells infected with HBV.** HBV cccDNA from HBV infected cells was purified via Hirt extraction method, and was subsequently detected by Southern blotting[20]. HBeAg levels were determined by chemiluminescent immunoassay using the HBeAg CLIA kit (Ig Biotechnology) and following the instructions from the manufacturer.

**Antibodies.** Primary antibodies used were anti-PCNA (for western blotting, 1:500 dilution, clone PC10, sc-56, lot# E2418, Santa Cruz Biotechnology, Dallas, TX), anti-PCNA (for immune-depletion, clone F2, sc-25280, lot #H1417, Santa Cruz Biotechnology), anti-POLD1 (for western blotting, 1:1000 dilution, rabbit polyclonal, 15646-1-AP, Proteintech, Rosemont, IL), anti-FEN-1 (for western blotting, 1:1500 dilution, rabbit polyclonal, A300-256A, Bethyl Laboratories, Montgomery, TX), anti-LIG1 (for western blotting, 1:1000 dilution, clone 10H5, GTX70141, lot#809703999, GeneTex, Irvine, CA), and anti-RFC4 (for western blotting, 1:1000 dilution, clone 1320, GTX70285, lot#14130, GeneTex).

**Statistical analyses.** We compared repair efficiencies from two groups of data in this study (e.g., NA-RrcDNA and RrcDNA; or drug treatments and mock treatments), and a t test is one of the most commonly used statistical analyses to compare two groups. Therefore, statistical analyses were performed by two-stage step-up t test method from Graphpad Prism version 7.0d. To control the false discovery rate, the discovery was determined using the two-stage linear step-up procedure of Benjamini, Krieger, and Yekutieli[44]. Each pair was analyzed individually, without assuming a consistent standard deviation (SD).

**Reporting summary.** Further information on research design is available in the Nature Research Reporting Summary linked to this article.

## Data availability

All data that support the findings of this study are available in the manuscript and the supplementary information files. Source data are provided with this paper.

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

## Acknowledgements
This work was supported in part by grants from the National Institutes of Health (R01 AI138797 and R01 AI153236 both to A.P.), a Research Scholar Award from the American Cancer Society (RSG-15-048-01-MPC to A.P.), a Burroughs Wellcome Fund Award for Investigators in Pathogenesis (to A.P.), a postdoctoral fellowship award from New Jersey Commission on Cancer Research (NJCCR) (DFHS17PPC011 to L.W.), and funding from Princeton University. We are grateful to P. Modrich (Duke University) for sharing RFC and POLδ expression plasmids, and to the Zakian lab for sharing equipment. We thank Jenna Gaska and members of the Ploss lab for providing critical feedback on this manuscript.

## Author contributions
The project was conceived by L.W. and A.P. All experiments were performed by L.W. All data were analyzed by L.W. and A.P. The manuscript was written by L.W. and A.P. Supplementary Fig. 8a was created by L.W. using BioRender licensed to Princeton University.

## Competing interests
The authors declare no competing interests.
