## [Peer Review File · Nature Communications]

Editorial Note: Parts of this Peer Review File have been redacted as indicated to remove third-party material where no permission to publish could be obtained. When text is deleted in rebuttals and referee reports, add “[redacted]” in that location.

Reviewers' Comments:

Reviewer #1:

Remarks to the Author:

I previously reviewed this manuscript for Nature Microbiology. The authors have submitted an extensively revised version that is significantly improved. The authors have been incredibly responsive to all concerns raised by reviewers and provide thoughtful responses and strengthening new data. They provide an elegant in vitro analysis of a complex process and give insights into ways that cccDNA might be formed. The manuscript has been improved by the review process and is a worthy contribution to the field.

Reviewer #2:

Remarks to the Author:

This is the revised version of a ms with the same title that had previously been submitted to Nature Microbiology and had been reviewed by three referees, including myself, and then has been transferred to Nature Communications.

Meanwhile the authors have clarified various issues, including by extensive additional experiments, in part to confirm reproducibility and allow for assessing statistical significance, in part to demonstrate the versatility of their in vitro system (e.g. by new order-of-addition experiments) and the physiological relevance of their findings (by employing pharmacological inhibitors of key factors FEN-1 and POL δ in a cell culture HBV infection system). In the text, the authors now properly emphasize potential limitations of their in vitro data for HBV cccDNA formation in vivo.

Hence overall an excellent study has even further been improved.

I have just a few minor points:

1. p4, l105: "... Pa can be elongated by PCNA-POL δ , which slows down as it reaches the 5' terminus of Pb and gradually displaces the RNA primer ..."

Is there previous evidence for such slowing down, e.g. from cellular DNA replication studies, which could be cited to support this point? See also point 2.

2. p4, l113 and Fig. 2d: "short-lived" intermediate extension products labeled by asterisks: Is the position in the gel compatible with "slowing down" at the 5' terminus of Pb, as suggested in 1.? Is there other evidence supporting the slowing down specifically at that site?

3. p4, l120, Extended Data Fig 3: "... Of note, prolonged incubation did not further increase cccDNA formation (Extended Data Fig. 3). The reason of why cccDNA formation does not reach completion remains to be determined but cannot simply be explained by exhaustion of repair factors

since we previously demonstrated that addition of a fresh dose of nuclear extract at 60 min only marginally increased repair. "

These remain puzzling observations - what should be different between "fresh" vs. "old" substrates? Rather than just stating that the reasons will have to be determined I suggest to include, very briefly, some of the potential reasons as forwarded in the rebuttal to reviewer #1, Fig. 2, Fig. 3.

4. Extended Data Fig. 5: Order-of-addition experiments. It is very clear that sequential addition, in whatever order, gives different results from simultaneous addition, namely little or no fully repaired Pa and Pb. What strikes me is that the band patterns for FEN-1 first vs. RFC/PCNA/POLD first are nearly superimposable, and neither goes to completion. It almost looks as if one of the two factors is completely inactive when alone, so the eventual outcome is only determined by the other; e.g. FEN-1 activity might be blocked by the Neutravidin "adduct" structures on the beads used for immobilization - could the authors comment?

5. Omission of factors

Is there precedent (again from cellular DNA replication) how flaps affect displacement synthesis? Is it easier for a DNA polymerase to "slip" under a flapped downstream 5' end (especially when this is RNA and forms a "super-stable" RNA-DNA hybrid) and continue synthesis, i.e. displace that strand? Or, to the contrary, is a flap an obstacle for a DNA polymerase? A few sentences on what is known might help readers to more easily grasp what the expected outcome is; see also 7.

6. p6, l77: "... Since 5' protein adduct retards FEN-1 activity²⁹, ..." - did ref. 29 make such a general statement? I could imagine that such retardation (presumably by steric hindrance) depends on HOW the protein is bound; if that linkage was through an extended flexible aa sequence there might be little hindrance? Please check whether using the current generalizing wording is appropriate.

7. Potential relevance of displacement synthesis in vivo?

Fig. 1 for reviewers shows indeed little displacement run-off synthesis; nonetheless, "extensive strand displacement does occur under certain conditions, such as when FEN-1 or LIG1 activity of level is limiting"; and same for the minus-strand.

As p8, l254 (Extended Data Fig. 10) suggest FEN-1 (and RFC) "as the least and most rate-limiting factors" - isn't that a similar situation which would promote "extensive strand displacement"? Please comment briefly why this is NOT a contradiction.

8. p8, l271: "...hepatocytes are mainly quiescent in the steady state, and targeted delivery of inhibitory molecules to the liver during short-term treatment may have minimal side effects. ..." Perhaps a more cautious statement would be: "targeted delivery of inhibitory molecules to the liver during short-term treatment may MINIMIZE side effects. ..." - or even more cautiously, as in the rebuttal: "... molecules inhibiting DNA replication may prove to not be overtly hepatotoxic. "

9. p9, l286 and labeling in Fig. 7: "TDP2 or other proteases ..." - TDP2 is a phosphodiesterase, not a protease, hence the wording should be adapted to make this clear.

10. p9, l289: "... since FEN-1 removes flap with a protein adduct very inefficiently, thus removal of HBV polymerase by TDP2 facilitates this process. "

a) see point 6: is the generalization of inefficient removal of flaps with attached protein justified?

b) decide on which article to use in front of "flap"::

FEN-1 removes a flap with a protein adduct inefficiently...

FEN-1 removes the flap with a protein adduct inefficiently... (referring to the authors own data)

FEN-1 removes flaps with a protein adduct inefficiently...

c) "...removal of HBV polymerase by TDP2 WOULD facilitate this process..." to emphasize the hypothetical character of the statement.

11. p34, l 890, legend to Extended Data Fig. 7: "Aphidocolin specifically inhibit repair..." - should read "inhibits repair"

Reviewer #3:

Remarks to the Author:

In this revised manuscript, the authors clarify and strengthen their results investigating the step-wise process involving repair factors PCNA, RCF, POL δ , FEN1, and LIG1 in formation of Hepatitis B Virus (HBV) cccDNA. Due to the role of cccDNA in HBV chronicity, data presented in this revised manuscript provides an avenue for investigating potential routes for a) understanding HBV chronicity and b) potentially generating HBV treatments. Specifically, authors included additional information in this revised manuscript that supported their conclusions including, showing purification of recombinant proteins, repeating experiments and providing statistical analysis, and providing potential translational impact by using an in-cell assay. They provide appropriate discussion for their findings and their overall impact. Overall, this manuscript will make an important contribution to the field.

Minor comments:

1. Authors do not reference Figure 5e in the text, where they are examining PCNA immune-depletion from human nuclear extracts impact on the Pb strand in the text. This figure is only referenced in regards to drawing attention to the overall impact and specifically, when highlighting the impact on the minus strand. In those regards, for line 161, do authors mean to reference Figure 5e-h?
2. Figure 5d-g. Could authors clarify if lanes 1-4 are for NA-RrcDNA and lanes 5-8 are for RrcDNA as formatted in their other figures? As currently formatted, it can be assumed as such (following the previous figures), but otherwise is unclear what the difference is between these blots as it is not stated in the legend either.
3. Lines 164 & 166, suggest authors reference Extended Figure 6d-e specifically
4. Lines 173 & 176, suggest authors reference Extended Figure 6f and 6e respectively
5. Line 206, suggest authors reference Extended Figure 6b-c specifically
6. Line 210, suggest authors reference Extended Figure 6d-e specifically
7. Figure 6. Letters e & l appear to be shifted into panels d & k, suggesting authors move these down so they more efficiently designate the appropriate panels.
8. Extended data figure 9: Line 928, authors state all experiments were stated twice. Line 929, authors state experiments in b were repeated 3 times. Suggest removing discrepancy in line 928 as updated experiments were repeated and old line contradicts new statement.
9. Authors do not include a methods section for statistical analysis. Suggest authors include a brief section about how the statistical data was generated and their rationale for using a two-step t-test over other analyses.
0. Extended data figure E10. Suggest the authors clarify in the results section that the starting concentrations of repair factors that stay "constant" are different. As stated, it can be interpreted that constant concentrations are the same between samples and this leads to confusing results when looking at the figure. At a glance, E10b suggests PCNA requires less (1/625) to efficiently produce cccDNA compared to FEN1 (1/25), even though 1/625 for PCNA is "2.5 nM vs 1/25 of FEN1 being "0.8nM. Perhaps include starting concentrations for each factor in the image for figure 10b itself or in the results section? Further, line 262 uses EC50 while Extended Figure 10c uses IC50, please correct this discrepancy.

We thank all the reviewers' and editor's efforts to evaluate our manuscript. In this document, we have addressed all the points raised and highlighted our responses in blue. We have modified the manuscript and supplementary information file accordingly, with the modified parts labeled in blue also.

REVIEWERS' COMMENTS

Reviewer #1 (Remarks to the Author):

I previously reviewed this manuscript for Nature Microbiology. The authors have submitted an extensively revised version that is significantly improved. The authors have been incredibly responsive to all concerns raised by reviewers and provide thoughtful responses and strengthening new data. They provide an elegant in vitro analysis of a complex process and give insights into ways that cccDNA might be formed. The manuscript has been improved by the review process and is a worthy contribution to the field.

We thank the reviewer for her/his positive comments.

Reviewer #2 (Remarks to the Author):

This is the revised version of a ms with the same title that had previously been submitted to Nature Microbiology and had been reviewed by three referees, including myself, and then has been transferred to Nature Communications. Meanwhile the authors have clarified various issues, including by extensive additional experiments, in part to confirm reproducibility and allow for assessing statistical significance, in part to demonstrate the versatility of their in vitro system (e.g. by new order-of-addition experiments) and the physiological relevance of their findings (by employing pharmacological inhibitors of key factors FEN-1 and POL6 in a cell culture HBV infection system). In the text, the authors now properly emphasize potential limitations of their in vitro data for HBV cccDNA formation in vivo.

Hence overall an excellent study has even further been improved.

We thank the reviewer for her/his positive comments.

I have just a few minor points:

1. p4, l105: "... Pa can be elongated by PCNA-POL6, which slows down as it reaches the 5' terminus of Pb and gradually displaces the RNA primer ..."

Is there previous evidence for such slowing down, e.g. from cellular DNA replication studies, which could be cited to support this point? See also point 2.

We thank the reviewer for her/his comments. We have added two references in lines 108 to support this point in the manuscript.

First, the slowing down of POL6 when it encounters duplex DNA has been previously reported in biochemical assays [Ganai, R. A., Zhang, X. P., Heyer, W. D. & Johansson, E. Strand displacement synthesis by yeast DNA polymerase epsilon. *Nucleic Acids Res.* 44, 8229-8240 (2016)]. In this study by Ganai *et. al*, Figure 5 (we modified this figure to Figure 1 for reviewers shown below) has

[redacted]

Figure 1 for reviewers.

Comparison of strand displacement synthesis between different polymerases. **a)** Substrate used in strand displacement synthesis by Pol c and Pol 6. This substrate contains a 5-nucleotide flap at the 5' end of the blocking DNA 22-mer primer. The 50-mer primer was radioactively labeled at the 5' end. Position 58nt indicates the end of the flap, position 59nt indicates one nucleotide into the duplex DNA. **b)** DNA substrates were mixed with DNA polymerases in a buffer containing dNTPs and magnesium acetate for the indicated times. #, a major pausing site for Pol 6 at the position one nucleotide after the flap.

shown yeast POL6 paused (slowed down) as it reached duplex DNA. As shown in Figure 1 for reviewers, a substrate containing a flap was used to evaluate whether Polc and Pol6 slow down as the polymerases encounter the flap and the duplex DNA. Since the radioactively labeled substrate used is very short (Figure 1a for reviewers), the extension by these polymerases can be determined in single nucleotide

resolution. The result showed that a prominent band (#) in the reaction containing POL6. The position of this band corresponds to the site one nucleotide into the duplex DNA. As FEN-1 prefers to cut the flap one nucleotide into the duplex DNA, the stalling of POL6 at this position leads to formation of a fragment that can be directly ligated with FEN-1 cleaved counterpart. This data is consistent with our observation and hypothesis.

In addition, another study [Stodala. J & Burger. P. Resolving individual steps of Okazaki fragment maturation at msec time-scale. *Nat Struct Mol Biol.* 23 (5), 402-408 (2016)] has specifically looked at the maturation of Okazaki fragments, and found that when POL6 encounters an RNA primer (this RNA primer is

engaged in "super-stable" RNA-DNA hybrid), its rate is reduced to 10-20% of that when no RNA-DNA duplex is present.

2. p4, I113 and Fig. 2d: "short-lived" intermediate extension products labeled by asterisks: Is the position in the gel compatible with "slowing down" at the 5' terminus of Pb, as suggested in 1.? Is there other evidence supporting the slowing down specifically at that site?

As our data suggested the extended Pa band (asterisks) can be directly ligated with FEN-1 processed Pb band (therefore being short-lived), the position of Pa band (asterisks) is most likely to be at one nucleotide into the duplex DNA as described in point 1. The single nucleotide resolution gel in point #1 supports this hypothesis.

3. p4, I120, Extended Data Fig 3: " ... Of note, prolonged incubation did not further increase cccDNA formation (Extended Data Fig. 3). The reason of why cccDNA formation does not reach completion remains to be determined but cannot simply be explained by exhaustion of repair factors since we previously demonstrated that addition of a fresh dose of nuclear extract at 60 min only marginally increased repair. "

These remain puzzling observations - what should be different between "fresh" vs. "old" substrates? Rather than just stating that the reasons will have to be determined I suggest to include, very briefly, some of the potential reasons as forwarded in the rebuttal to reviewer #1, Fig. 2, Fig. 3.

We thank the reviewer for her/his comments. We added the following statement: 'Possible reasons include: 1) some repair events are reversible and the reaction reaches equilibrium; 2) some nucleases may revert cccDNA to rcDNA; 3) formation of dead-end repair intermediates that are refractory for further processing into cccDNA'.

4. Extended Data Fig. 5: Order-of-addition experiments. It is very clear that sequential addition, in whatever order, gives different results from simultaneous addition, namely little or no fully repaired Pa and Pb. What strikes me is that the band patterns for FEN-1 first vs. RFC/PCNA/POLD first are nearly superimposable, and neither goes to completion. It almost looks as if one of the two factors is completely inactive when alone, so the eventual outcome is only determined by the other; e.g. FEN-1 activity might be blocked by the Neutravidin "adduct" structures on the beads used for immobilization - could the authors comment?

We thank the reviewer for her/his comments. Neutravidin "adduct" structures on the beads used for immobilization are only on the minus strands, and only the repair of the minus strand is inhibited. However, we have shown that Neutravidin does not affect the repair of the plus strand (Fig. 2d-e). Therefore, we only examined the repair of the plus strand (Pa and Pb) in Extended Data Figure. 5.

As suggested by the review's careful analysis, our data indicate that the successful repair of Pa and Pb needs the cooperation of FEN-1 and RFC/PCNA/POLD. The full repair of the plus strand requires the extension of Pa by RFC/PCNA/POLD, and FEN-1 cleavage of Pb at a specific location that can perfectly align with Pa, so that these two fragments can be ligated by LIG1. It is most likely that the FEN-1 mediated cleavage of Pb at the correct location is regulated by the presence of RFC/PCNA/POLD. FEN-1 has been shown to be able to interact with PCNA, this interaction may play a critical role in the coordination of Pa and Pb processing. In this case, FEN-1 first vs. RFC/PCNA/POLD first will both produce misaligned Pa and Pb, which could not be ligated, and RFC/PCNA/POLD will keep extending Pa until run-off. The presence of run-off Pa is evident in Extended Data Fig. 5b and supports this hypothesis, which could explain why the band patterns for FEN-1 first vs. RFC/PCNA/POLD first are nearly superimposable, since the optimal functions of these two sets of factors are linked.

To further support this hypothesis, a study [Stodala. J & Burger. P. Resolving individual steps of Okazaki fragment maturation at msec time-scale. *Nat Struct Mol Biol.* 23 (5), 402-408 (2016)] has specifically looked at the maturation of Okazaki fragment-like structures, and showed that PCNA / POL6 / FEN-1 cooperated with each other to remove RNA primers.

5. Omission of factors

Is there precedent (again from cellular DNA replication) how flaps affect displacement synthesis? Is it easier for a DNA polymerase to "slip" under a flapped downstream 5' end (especially when this is RNA and forms a "super-stable" RNA-DNA hybrid) and continue synthesis, i.e. displace that strand? Or, to the contrary, is a flap an obstacle for a DNA polymerase? A few sentences on what is known might help readers to more easily grasp what the expected outcome is; see also 7.

We thank the reviewer for her/his comments. The study we showed in point #1 (Figure 1 for reviewers) have indicated that Pol 6 can pass the flap and slow down by the DNA duplex during displacement synthesis.

Consistent with this, another study [Stodala. J & Burger. P. Resolving individual steps of Okazaki fragment maturation at msec time-scale. *Nat Struct Mol Biol.* **23** (5), 402-408 (2016)] has specifically looked at the maturation of Okazaki fragment-like structures, and found that when POL 6 encounters an RNA primer (this RNA is engaged in "super-stable" RNA-DNA hybrid), its rate is reduced to 10-20% of that when no RNA-DNA duplex is present. Albeit being slowed down, POL6 can slip under this RNA and displace about 1-2 nucleotides, which are cleaved by FEN-1. Then POL 6 continues displacing about 1-2 nucleotides, which are again cleaved by FEN-1, this iterative displacement and cleavage cycles mediated by PCNA / POL6 / FEN-1 will lead to complete removal of the

RNA primer.

We have referenced these two studies in lines 107-108, and added the following sentence in lines 150-152 to help readers to more easily grasp what the expected outcome is: 'Consistent with our observations, a previous study showed that PCNA /POL δ / FEN-1 cooperatively mediates iterative RNA primer displacement and cleavage cycles in Okazaki-fragments, which leads to its complete removal'.

6. p6, l77: " ... Since 5' protein adduct retards FEN-1 activity³², ... " - did ref. 32 make such a general statement? I could imagine that such retardation (presumably by steric hindrance) depends on HOW the protein is bound; if that linkage was through an extended flexible aa sequence there might be little hindrance? Please check whether using the current generalizing wording is appropriate.

We thank the reviewer for her/his comments. Ref. 32 showed that Biotin-Streptavidin type of DNA-protein adducts (similar to our Biotin-NeutrAvidin) inhibit FEN-1 activity. To avoid current generalizing wording, we have modified this sentence to 'Since 5' protein adduct (such as DNA-Biotin-Streptavidin) reduces FEN-1 activity³²'.

7. Potential relevance of displacement synthesis in vivo?

Fig. 1 for reviewers shows indeed little displacement run-off synthesis; nonetheless, "extensive strand displacement does occur under certain conditions, such as when FEN-1 or LIG1 activity of level is limiting"; and same for the minus-strand.

As p8, l254 (Extended Data Fig. 10) suggest FEN-1 (and RFC) "as the least and most rate-limiting factors" - isn't that a similar situation which would promote "extensive strand displacement"? Please comment briefly why this is NOT a contradiction.

We thank the reviewer for her/his comments. We think 'Extensive strand displacement' mostly leads to run-off and linearization of rcDNA (longer than 1.0 genome length due to the flap on the minus strand is not removed when FEN-1 is limiting). To generate a functional cccDNA molecule, this flap (r sequence) must be removed; otherwise it will interfere with the replication of progeny rcDNA. Therefore, although circularization of this over-length linearized rcDNA via error prone non-homologous end joining could theoretically lead to formation of cccDNA-like molecule, this over-length molecule would be defective.

In the previous response to reviewers, with respect to the sentence "extensive strand displacement does occur under certain conditions, such as when FEN-1 or LIG1 activity of level is limiting", we meant that this extensive strand displacement can happen, but is likely detrimental to formation of functional cccDNA molecule.

With respect to the sentence 'As p8, I254 (Extended Data Fig. 10) suggest FEN-1 (and RFC) "as the least (and most) rate-limiting factors"', we meant that in cells, FEN-1 is likely abundant enough (promote cleavage of flaps) while RFC is likely to be limiting (restraining strand extension), which would lead to a situation that disfavors extensive strand displacement. Therefore, it is consistent with the notion that extensive strand displacement most likely does not occur often.

8.p8, I279: "...hepatocytes are mainly quiescent in the steady state, and targeted delivery of inhibitory molecules to the liver during short-term treatment may have minimal side effects. ..."

Perhaps a more cautious statement would be: "targeted delivery of inhibitory molecules to the liver during short-term treatment may MINIMIZE side effects. ..." - or even more cautiously, as in the rebuttal: "...molecules inhibiting DNA replication may prove to not be overtly hepatotoxic. "

We thank the reviewer for her/his comments. We have changed this sentence to "targeted delivery of inhibitory molecules to the liver during short-term treatment may minimize side effects. ...", as suggested by the reviewer.

9.p9, I286 and labeling in Fig. 7: "TDP2 or other proteases ..." - TDP2 is a phosphodiesterase, not a protease, hence the wording should be adapted to make this clear.

We thank the reviewer for her/his comments. We have deleted 'other' in both the text and the labeling in Fig. 7.

0. p9, I289: "... since FEN-1 removes flap with a protein adduct very inefficiently, thus removal of HBV polymerase by TDP2 facilitates this process. " a)see point 6: is the generalization of inefficient removal of flaps with attached protein justified?

We modified this sentence to 'since FEN-1 removes the flap with a protein adduct (biotin-NeutrAvidin) very inefficiently (Fig. 2h, Extended Data Fig. 4d-e), thus removal of HBV polymerase by TDP2 would facilitate this process', to emphasize that it may not be the case for TDP2. This also addresses point c) below.

b)decide on which article to use in front of "flap"::

FEN-1 removes a flap with a protein adduct inefficiently...

FEN-1 removes the flap with a protein adduct inefficiently... (referring to the authors own data)

FEN-1 removes flaps with a protien adduct inefficiently...

We change the sentence to 'FEN-1 removes the flap with a protein adduct (biotin-NeutrAvidin) very inefficiently (Fig. 2h, Extended Data Fig. 4d-e)' as suggested by the reviewer.

c) ".removal of HBV polymerase by TDP2 WOULD facilitate this process..." to emphasize the hypothetical character of the statement.

We modified this sentence as suggested by the reviewer.

11. p34, l 890, legend to Extended Data Fig. 7: " Aphidocolin specifically inhibit repair..." - should read "inhibits repair"

We modified this sentence as suggested by the reviewer.

Reviewer #3 (Remarks to the Author):

In this revised manuscript, the authors clarify and strengthen their results investigating the step-wise process involving repair factors PCNA, RCF, POL δ , FEN1, and LIG1 in formation of Hepatitis B Virus (HBV) cccDNA. Due to the role of cccDNA in HBV chronicity, data presented in this revised manuscript provides an avenue for investigating potential routes for a) understanding HBV chronicity and b) potentially generating HBV treatments. Specifically, authors included additional information in this revised manuscript that supported their conclusions including, showing purification of recombinant proteins, repeating experiments and providing statistical analysis, and providing potential translational impact by using an in-cell assay. They provide appropriate discussion for their findings and their overall impact. Overall, this manuscript will make an important contribution to the field.

We thank the reviewer for her/his positive comments.

Minor comments:

1. Authors do not reference Figure 5e in the text, where they are examining PCNA immune-depletion from human nuclear extracts impact on the Pb strand in the text. This figure is only referenced in regards to drawing attention to the overall impact and specifically, when highlighting the impact on the minus strand. In those regards, for line 161, do authors mean to reference Figure 5e-h?

We thank the reviewer for his/her comments. We have added a line in #165 'impaired repair of the Pb fragment of the plus strand' to reference Figure 5e, and have corrected 'Figure 5d-h' to 'Figure 5e-h' in line #166.

2. Figure 5d-g. Could authors clarify if lanes 1-4 are for NA-RrcDNA and lanes 58 are for RrcDNA as formatted in their other figures? As currently formatted, it

can be assumed as such (following the previous figures), but otherwise is unclear what the difference is between these blots as it is not stated in the legend either.

We thank the reviewer for his/her comments. We have added NA-RrcDNA and RrcDNA labels to Figure 5d-g and the figure legend.

3. Lines 164 & 166, suggest authors reference Extended Figure 6d-e specifically

We thank the reviewer for his/her comments. We have referenced Extended Figure 6d-e specifically as suggested by the reviewer.

4. Lines 173 & 176, suggest authors reference Extended Figure 6f and 6g respectively

We have referenced Extended Figure 6f-g specifically as suggested by the reviewer.

5. Line 206, suggest authors reference Extended Figure 6b-c specifically

We have referenced Extended Figure 6b-c specifically as suggested by the reviewer.

6. Line 210, suggest authors reference Extended Figure 6d-e specifically

We have referenced Extended Figure 6d-e specifically as suggested by the reviewer.

7. Figure 6. Letters e & l appear to be shifted into panels d & k, suggesting authors move these down so they more efficiently designate the appropriate panels.

We thank the reviewer for his/her comments. We have moved panel label e & i, d & k down in Figure 6.

8. Extended data figure 9: Line 928, authors state all experiments were stated twice. Line 929, authors state experiments in b were repeated 3 times. Suggest removing discrepancy in line 928 as updated experiments were repeated and old line contradicts new statement.

We thank the reviewer for his/her comments. We have removed the sentence 'all experiments were stated twice' as suggested.

9. Authors do not include a methods section for statistical analysis. Suggest authors include a brief section about how the statistical data was generated and their rationale for using a two-step t-test over other analyses.

We thank the reviewer for his/her comments. We have added a 'Statistical analyses' section in the Methods to explain about how the statistical data was generated and the rationale for using a two-step t-test.

10. Extended data figure E10. Suggest the authors clarify in the results section that the starting concentrations of repair factors that stay "constant" are different. As stated, it can be interpreted that constant concentrations are the same between samples and this leads to confusing results when looking at the figure. At a glance, E10b suggests PCNA requires less (1/625) to efficiently produce cccDNA compared to FEN1 (1/25), even though 1/625 for PCNA is ~2.5 nM vs 1/25 of FEN1 being ~0.8nM. Perhaps include starting concentrations for each factor in the image for figure 10b itself or in the results section? Further, line 262 uses EC50 while Extended Figure 10c uses IC50, please correct this discrepancy.

We thank the reviewer for his/her comments. As suggested by the reviewer, we have changed the sentence in question to 'We next aimed to determine the minimal concentration of each factor needed to catalyze the reaction by serially diluting one factor while keeping the other four factors' concentrations the same as their starting concentrations (1.5 μ M PCNA, 35 nM RFC, 20 nM POL δ , 100 nM LIG1, and 20 nM FEN-1).'

We have also changed 'IC50' in Extended Figure. 10c to 'EC50'.